# Proximal Policy Gradient Arborescence for Quality Diversity Reinforcement Learning

**Sumeet Batra**
University of Southern California
Los Angeles, CA 90089
ssbatra@usc.edu

**Bryon Tjanaka**
University of Southern California
Los Angeles, CA 90089
tjanaka@usc.edu

**Matthew C. Fontaine**
University of Southern California
Los Angeles, CA 90089
mfontain@usc.edu

**Aleksei Petrenko**
University of Southern California
Los Angeles, CA 90089
petrenko@usc.edu

**Stefanos Nikolaidis**
University of Southern California
Los Angeles, CA 90089
nikolaid@usc.edu

**Gaurav S. Sukhatme**
University of Southern California
Los Angeles, CA 90089
gaurav@usc.edu

## Abstract

Training generally capable agents that thoroughly explore their environment and learn new and diverse skills is a long-term goal of robot learning. Quality Diversity Reinforcement Learning (QD-RL) is an emerging research area that blends the best aspects of both fields – Quality Diversity (QD) provides a principled form of exploration and produces collections of behaviorally diverse agents, while Reinforcement Learning (RL) provides a powerful performance improvement operator enabling generalization across tasks and dynamic environments. Existing QD-RL approaches have been constrained to sample efficient, deterministic *off-policy* RL algorithms and/or evolution strategies, and struggle with highly stochastic environments. In this work, we, for the first time, adapt on-policy RL, specifically Proximal Policy Optimization (PPO), to the Differentiable Quality Diversity (DQD) framework and propose additional improvements over prior work that enable efficient optimization and discovery of novel skills on challenging locomotion tasks. Our new algorithm, Proximal Policy Gradient Arborescence (PPGA), achieves state-of-the-art results, including a 4x improvement in best reward over baselines on the challenging humanoid domain.

## 1 Introduction

Quality Diversity (QD) algorithms enable the exploration and discovery of diverse skills in a behavior space. For example, a QD algorithm can train different locomotion gaits for a walker (Cully et al., 2015), discover different grasping trajectories for a manipulator (Morel et al., 2022), or generate a diverse range of human faces (Fontaine & Nikolaidis, 2021). However, since these algorithms are generally oriented towards solving exploration problems, they struggle to find performant policies in high-dimensional robot learning tasks. QD-RL is an emerging field that attempts to combine the principled exploration capabilities of QD with the powerful performance improvement capabilities of RL. Prior methods have leveraged *off-policy* RL, specifically TD3, to estimate the gradient of performance, and either Evolution Strategies (ES) or TD3 to estimate the gradient of diversity in order to search for diverse, high-quality policies. They have shown success in exploration problems and certain robot locomotion tasks (Nilsson & Cully, 2021; Pierrot et al., 2022; Tjanaka et al., 2022b). Nonetheless, there remains a gap in performance between QD-RL and standard RL algorithms on continuous control tasks. Furthermore, off-policy RL algorithms were not designed with massive parallelization in mind, and there is little literature that explores how to leverage modern massively-

parallelized simulators with these algorithms, whereas there are numerous works exploring on-policy RL in these regimes (Makoviychuk et al., 2021; Rudin et al., 2021; Handa et al., 2022; Batra et al., 2021; Huang et al., 2022).

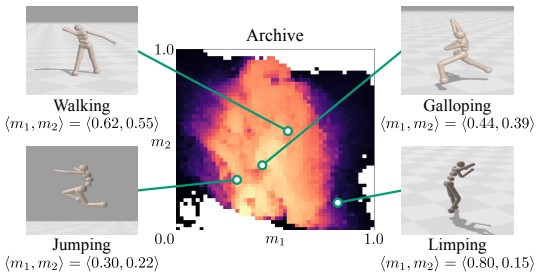

Figure 1: PPGA finds a diverse archive of high-performing locomotion behaviors for a humanoid agent by combining PPO gradient approximations with Differentiable Quality Diversity algorithms. The archive's dimensions correspond to the measures $m_1$ and $m_2$, i.e., the proportion of time that the left and right feet contact the ground. The color of each cell shows the objective value, i.e., how fast the humanoid moves. For instance, jumping moves the humanoid forward quickly, with the left and right feet individually contacting the ground 30% and 22% of the time, respectively.

From our investigation of prior methods, simply combining existing QD methods with an RL algorithm tends not to scale well to high-dimensional, highly dynamical systems such as Humanoid. For example, all QD-RL algorithms for locomotion to date use non-Markovian measures of behavioral diversity, which in many cases prevents direct RL-optimization. Most algorithms instead opt for policy parameter mutation, which struggles to scale well with deep neural networks. Prior methods that investigated combining Differentiable Quality Diversity and *off-policy* RL (Tjanaka et al., 2022b) achieved similar results as other baselines. However, given the gap in performance between standard RL and QD-RL algorithms in terms of best-performing policy, we believe that DQD algorithms, under a different formulation more synergistic with its underlying mechanisms, can close this gap. To this end, we leverage Proximal Policy Optimization (PPO) (Schulman et al., 2017), a popular *on-policy* RL algorithm, with Differentiable Quality Diversity (DQD) (Fontaine & Nikolaidis, 2021) because of the already present synergy. Specifically, DQD algorithms CMA-MEGA (Fontaine & Nikolaidis, 2021), and its more recent variation CMA-MAEGA (Fontaine & Nikolaidis, 2023), maintain a single search point (or policy in the case of RL) that moves through the behavior space and fills in new, unexplored regions with offspring policies constructed via gradient information *collected from online data*. It is through this high level view that we see the emergent synergy between PPO and DQD, in that PPO can be used to collect gradient estimates from online data when one or both of the objective and measure functions are Markovian and non-differentiable.

We make several key changes to CMA-MAEGA and PPO to maximally leverage their synergy. Our new algorithm, Proximal Policy Gradient Arborescence (PPGA), to the best of our knowledge, is the first QD-RL algorithm to not only achieve 4x performance in best reward on the humanoid domain, but achieve the *same* level of performance as PPO without sacrificing *any* of the diversity in the discovered policies. Specifically, we make the following contributions:

**(1)** We propose a vectorized implementation of PPO, VPPO, that jointly computes the objective and measure gradients with little overhead and without running separate PPO instances for each task **(2)** We generalize prior *CMA*-based DQD algorithms as instances of Natural Evolution Strategies (NES) and show that contemporary NES methods, specifically xNES, enable better training stability and performance for DQD algorithms **(3)** We introduce the notion of Markovian Measure Proxies (MMPs), which makes the typically non-Markovian measure functions used in QD-RL amenable to RL-optimization **(4)** We propose a new method to move the current search point, hereon referred to as the "search policy", to unexplored regions of the archive by iteratively "walking" it using collected online data and RL optimization of a novel multi-objective reward function.

## 2 BACKGROUND

### 2.1 DEEP REINFORCEMENT LEARNING

**Reinforcement Learning** algorithms search for a policy, a mapping of states to actions, that maximizes cumulative reward in an environment. RL assumes the discrete-time Markov Decision Process (MDP) formalism $(\mathcal{S}, \mathcal{A}, \mathcal{R}, \mathcal{P}, \gamma)$ where $\mathcal{S}$ and $\mathcal{A}$ are the state and action spaces respectively, $\mathcal{R}(s, a)$ is the reward function, $\mathcal{P}(s'|s, a)$ defines state transition probabilities, and $\gamma$ is the discount factor. The RL objective is to maximize the discounted episodic return of a policy $\mathbb{E}\left[\sum_{k=0}^{T-1} \gamma^k R(s_k, a_k)\right]$ where $T$ is episode length. **Deep Reinforcement Learning** solves the RL problem by finding a policy $\pi_\theta(a_t|s_t)$ parameterized by a deep neural network $\theta$ that represents a state-action mapping.

**On-policy Deep RL** methods directly learn the policy $\pi_\theta$ using experience collected by that policy or a recent version thereof. Contemporary methods (Mnih et al., 2016) fit the value function $V_\phi(s_t)$ to discounted returns and estimate the advantage $\hat{A}_t = \sum_{k=t}^{T-1} \gamma^k R(s_k, a_k) - V_\phi(s_t)$, which corresponds to the value of an action over the current policy (Schulman et al., 2016). From here, the gradient of the objective w.r.t. $\theta$, or **policy gradient**, can be estimated as $\hat{\mathbb{E}}_t\left[\nabla_\theta \log \pi_\theta(a_t|s_t)\hat{A}_t\right]$, and the policy $\pi_\theta$ is trained using mini-batch gradient descent.

**Trust region** policy gradient Deep RL methods constrain the policy updates to maintain the proximity of $\pi_\theta$ to the *behavior* policy $\pi_{\theta_{old}}$ that was used to collect the experience. TRPO (Schulman et al., 2015) takes the largest policy improvement step that satisfies the strict KL-divergence constraint. Proximal Policy Optimization (PPO) (Schulman et al., 2017) approximates the trust region by optimizing a clipped surrogate objective where $r_t(\theta) = \frac{\pi_\theta(a_t|s_t)}{\pi_{\theta_{old}}(a_t|s_t)}$ is the importance sampling ratio:

$$L(\theta) = \hat{\mathbb{E}}_{\pi_\theta}\left[\min(r_t(\theta)\hat{A}_t), \text{clip}(r_t(\theta), 1 - \epsilon, 1 + \epsilon)\hat{A}_t\right].$$

**Off-policy Deep RL algorithms** learn parameterized state-action value functions $Q_\theta(s_t, a_t)$ that estimate the value of taking action $a_t$ in state $s_t$. Then, actions are taken with a greedy policy $\arg\max_a Q_\theta(s_t, a_t)$, or an $\varepsilon$-greedy variation thereof. Q-functions can be learned from experience collected by recent or past versions of the policy or another policy altogether.

In continuous control problems, it can be difficult to find $a^* = \arg\max_a Q_\theta(s_t, a_t)$ due to an infinite number of possible actions. To work around this issue, off-policy methods such as DDPG (Lillicrap et al., 2016) and TD3 (Fujimoto et al., 2018) learn a deterministic policy $\mu_\phi(s_t)$ by solving $max_\phi(Q_\theta(s_t, \mu_\phi(s_t))$ using gradient ascent. Other off-policy methods, such as soft actor-critic (SAC) Haarnoja et al. (2018), maintain an explicit policy $\pi_\theta$, but similarly derive the policy gradient from the critic, allowing them to learn from off-policy data as well.

### 2.2 QUALITY DIVERSITY OPTIMIZATION

Unlike single-objective optimization methods such as RL, Quality Diversity algorithms search for an archive of high-performing, diverse policies. An optimal archive essentially answers the question, "how does performance change with behavior?" by mapping out the optimization landscape of a pre-defined behavior space. The QD problem (Chatzilygeroudis et al., 2021) assumes an *objective* function $f(\cdot)$ that quantifies the agent's performance and $k$ *measure* functions $m_1(\cdot)...m_k(\cdot)$ that characterize the agent's behavior. The measure functions, represented jointly as $\mathbf{m}(\cdot)$, define an embedding the QD algorithm should span with diverse policies. The **QD objective** is to find a policy that maximizes $f$ for every possible output of $\mathbf{m}$. However, the embedding formed by $\mathbf{m}$ is continuous, so the embedding space is discretized into a tessellation of $M$ cells. The QD objective then becomes to maximize $\sum_{i=1}^{M} f(\theta_i)$, where $\theta_i$ is a policy whose measures $\mathbf{m}(\theta_i)$ fall in cell $i$ of the tesselation.

QD algorithms originated with NSLC (Lehman & Stanley, 2011a;b) and MAP-Elites (Mouret & Clune, 2015; Cully et al., 2015). While these early QD algorithms built on genetic algorithms, modern QD algorithms incorporate optimization techniques like evolution strategies (Fontaine et al., 2020; Conti et al., 2018; Colas et al., 2020), gradient ascent (Fontaine & Nikolaidis, 2021; 2023), and differential evolution (Choi & Togelius, 2021). Several works have applied QD optimization to generative design (Hagg et al., 2020; Gaier et al., 2018), procedural content generation (Gravina

et al., 2019; Earle et al., 2022; Khalifa et al., 2018), robot manipulation (Morrison et al., 2020), and reinforcement learning (Nilsson & Cully, 2021; Tjanaka et al., 2022b; Pierrot & Flajolet, 2023).

## 2.3 DIFFERENTIABLE QUALITY DIVERSITY

The Differentiable Quality Diversity (DQD) (Fontaine & Nikolaidis, 2021) algorithm Covariance Matrix Adaptation Map Elites via Gradient Arborescence (CMA-MEGA) considers the first-order QD problem where the objective and measure functions are differentiable, with gradients w.r.t. policy parameters represented as $\nabla f = \frac{\partial f}{\partial \theta}$ and $\nabla \mathbf{m} = \left[ \frac{\partial m_1}{\partial \theta}, ..., \frac{\partial m_k}{\partial \theta} \right]$. CMA-MEGA maintains a *search policy* $\pi_{\theta_\mu}$ in policy parameter space ($\theta_\mu \in \mathbb{R}^N$) corresponding to some cell in the archive given by the measures $< m_1(\pi_{\theta_\mu}), ..., m_k(\pi_{\theta_\mu}) >$, and a search distribution in objective-measure gradient coefficient space maintained by CMA-ES (Hansen, 2016), a zeroth-order optimizer that optimizes the coefficient distribution to produce coefficient vectors that point in the direction of *greatest archive improvement*. At a high level, CMA-MEGA branches off policies from the search policy in order to locally fill the archive, and then steps the search policy to new, unexplored regions of the archive. During the branching step, the gradients $< \nabla f, \nabla \mathbf{m} >_{\theta_\mu}$ and $\lambda$ gradient coefficient vectors $< c_0, ..., c_k >_1, ..., < c_0, ..., c_k >_\lambda$ sampled from the CMA-ES search distribution $\mathbf{c}_i \sim \mathcal{N}(\mu, \Sigma) \in \mathbb{R}^{k+1}$ are combined via the dot product i.e. $< \nabla f, \nabla \mathbf{m} >_{\theta_\mu} \cdot \mathbf{c}_1, ...$ to produce local gradients $\nabla_1, ..., \nabla_\lambda$ around the search policy. Applying the gradients to the search policy gives us $\lambda$ branched policies $\pi_{\theta_1}, ..., \pi_{\theta_\lambda}$. The new policies can then be ranked by how much they improve the archive, i.e., $f(\pi_{\theta_i}) - f(\pi_{\theta_{old}}), i \in [1, \lambda]$, where $\pi_{\theta_{old}}$ is the incumbent policy in the archive corresponding to the same cell as $\pi_{\theta_i}$. Branched policies that map to new, unexplored cells in the archive have $f(\pi_{\theta_{old}})$ set to some minimum threshold. This implicitly biases the ranking towards the exploration of new, unvisited cells. This ranking is given to CMA-ES, which internally performs an update that steps the search distribution in the direction of the natural gradient w.r.t. greatest archive improvement. CMA-ES returns weights $w_1, ..., w_\lambda$ such that $\nabla_{step} = < w_1, ..., w_\lambda > \cdot < \nabla_1, ..., \nabla_\lambda >$ is the natural gradient in parameter space. This weighted linear recombination of the branching gradients is then used to step the search policy in the direction of greatest archive improvement $\theta_\mu \leftarrow \theta_\mu + \alpha \nabla_{step}$.

The current state-of-the-art DQD algorithm, Covariance Matrix Adaptation Map Annealing via Gradient Arborescence (CMA-MAEGA) (Fontaine & Nikolaidis, 2023), introduced the concept of soft archives to CMA-MEGA. Instead of maintaining the best policy in each cell, the archive maintains a threshold $t_e$ and updates the threshold by $t_e \leftarrow (1 - \alpha)t_e + \alpha f(\pi_{\theta_i})$ when a new policy $\pi_{\theta_i}$ crosses the threshold of its cell $e$. The hyperparameter $0 \leq \alpha \leq 1$, referred to as the *archive learning rate*, controls how much time is spent optimizing a region of the archive before exploring a new region. Soft archives have many theoretical and practical benefits discussed in prior work (Fontaine & Nikolaidis, 2023). Our proposed PPGA algorithm builds directly on CMA-MAEGA.

## 2.4 QUALITY DIVERSITY REINFORCEMENT LEARNING

Unlike the standard DQD formulation in which the analytical gradients of $f$ and $\mathbf{m}$ can be computed, the QD-RL setting considers MDPs in which these functions are non-differentiable and must be *approximated* with model-free RL. The gradient approximations of $f$ and $\mathbf{m}$ can be used to improve the performance and diversity of agents in an archive. QD-RL methods can be roughly divided into two subgroups. The first set of approaches directly optimizes over the entire archive by sampling existing policies in the archive and applying operations to the policies' parameters that either improve their performance or diversity. For example, PGA-ME (Nilsson & Cully, 2021) collects experience from evaluated agents into a replay buffer and uses TD3 to derive a policy gradient that improves the performance of randomly sampled agents from the archive, while using genetic variation (Vassiliades & Mouret, 2018) on the same set of agents to improve diversity and fill new, unexplored cells. Similarly, QDPG (Pierrot et al., 2022) derives a policy and diversity gradient using TD3 and applies these operators to randomly sampled agents in the archive.

Whereas the first family of QD-RL algorithms simultaneously search the behavioral embedding in many different regions at once, the second family uses the DQD formulation i.e., maintains a *single search policy* that explores new local regions one at a time using objective-measure gradient *approximations*. In prior work (Tjanaka et al., 2022b), the authors considered objective gradient approximations via TD3 and OpenAI-ES, while approximating the measure function gradients with

OpenAI-ES. In this work, we notice the unique on-policy nature of DQD algorithms and present a novel formulation that exploits their synergy with PPO.

## 3 PROPOSED METHOD: THE PROXIMAL POLICY GRADIENT ARBORESCENCE ALGORITHM

We begin with the DQD algorithm CMA-MAEGA as our foundation. The algorithm can be roughly divided into three phases: (1) computing the objective-measure gradients for the branching phase, (2) providing the relative ranking of each branched policy w.r.t. the QD objective to CMA-ES, and (3) stepping the search policy in the direction of greatest archive improvement. Sections 3.1 and 3.2 focus on enabling RL optimization for phase one, 3.3 explores the connection between CMA-ES and NES and how this can improve training stability in phase two, and section 3.4 describes our method for walking the search policy with PPO in phase three.

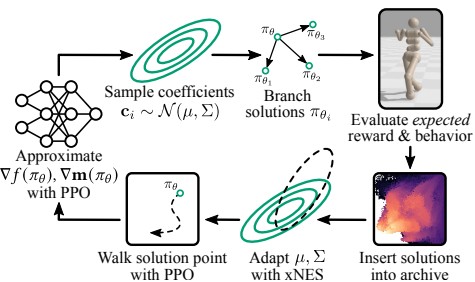

Figure 2: PPGA estimates $\nabla f, \nabla \mathbf{m}$ with PPO. We randomly sample gradient coefficients $\mathbf{c}$ and perform weighted linear recombination of the objective-measure gradients with $\mathbf{c}$ as the weights. This produces a population of gradients that, in turn, result in a population of branched policies. The policies are evaluated and inserted into the archive. xNES adapts the gradient coefficient distribution based on these insertions towards maximal archive improvement. The new mean of the coefficient distribution is used to walk the search policy towards a new, potentially unexplored region of the archive.

### 3.1 MARKOVIAN MEASURE PROXIES

QD problems often contain non-Markovian measure functions. For robot locomotion tasks, the standard measure function is the proportional foot contact time with the ground $m_i(\theta) = \frac{1}{T}\sum_{t=0}^{T}\delta_i(s_t)$ for each leg $i, i = 1...k$, where the Kronecker delta $\delta_i(s_t)$ indicates whether the $i$'th leg is in contact with the ground or not in state $s_t$, and $T$ is the episode length. However, this measure function is defined on a trajectory (i.e., the whole episode), making it non-Markovian and thus preventing us from using RL to estimate its gradient. To solve this issue, we introduce the notion of a **Markovian Measure Proxy** (MMP), which is a surrogate function that obeys the Markov property and has a positive correlation with the original measure function. For locomotion tasks, we can construct an MMP by simply removing the dependency on the trajectory and making the original measure function state-dependent, i.e., setting it to be $\delta_i(s_t)$. We can then use the exact same MDP as the standard RL formulation and replace the reward function with $\delta_i(s_t)$.

### 3.2 POLICY GRADIENTS FOR DIFFERENTIABLE QUALITY DIVERSITY OPTIMIZATION

PPO is an attractive choice as our objective-measure gradient estimator because of its ability to scale with additional parallel environments. Being an approximate trust region method, the constrained policy update step provides some robustness to noisy and non-stationary objectives. This is particularly important in the QD-RL setting, where the QD-objective is highly non-stationary – that is, the QD-objective changes with the state of the archive, which is updated on each QD iteration.

We treat the RL objective and $k$ MMPs, each one optimized by an actor-critic pair, as reward functions to optimize. Rather than spawning a new PPO instance with a separate actor and critic network for the RL objective $f$ and each MMP $\delta_i(s_t)$ independently, we start with a single actor $\pi_{\theta_\mu}(a|s)$ parameterized by the policy parameters $\theta_\mu$ of the current search policy, and $k + 1$ value functions $V_{\phi_f}, V_{\phi_{\delta_1}}, ..., V_{\phi_{\delta_k}}$. The actor is replicated $k + 1$ times, each one paired with a corresponding value function. The actors are combined into a single *vectorized policy* $\pi_{<\theta_\mu^1,...,\theta_\mu^{k+1}>}(a|s)$ that jointly optimizes $< f, \delta_1(s_t), ..., \delta_1(s_t) >$ for $N_1$ iterations, where $N_1$ is a configurable hyperparameter. We additionally modify the computation of the policy gradient into a *batched policy gradient* method, where intermediate gradient estimates of each function w.r.t. policy params only flow back to

the parameters corresponding to respective individual policies during minibatch gradient descent. After $N_1$ iterations, we separate the vectorized policy, giving a set of subpolicies with optimized parameters $\pi_{\theta_f}(a|s), ..., \pi_{\theta_{\delta_k}}(a|s)$ that perform better w.r.t. their objectives. In the case of measure functions where $m_i(\cdot)$ is the proportion foot contact time of the $i'th$ leg, $m_i(\pi_{\theta_{\delta_k(s_t)}}) > m_i(\pi_{\theta_\mu})$ i.e. the resulting policy will have a higher proportion foot contact time over the starting policy after optimization. Subtracting the initial parameters $(\theta_\mu)$ from each resulting policy gives us the desired objective-measure Jacobian $\left[\frac{\partial f}{\partial \theta_\mu}, \frac{\partial \delta_1}{\partial \theta_\mu}, ..., \frac{\partial \delta_k}{\partial \theta_\mu}\right]$, which can be linearly recombined in various ways to branch policies from $\theta_\mu$.

In addition to the VPPO implementation, we introduce the option to make the learnable action standard deviation parameter static. In the typical case, PPO decays this parameter over time in order to converge to a quasi-deterministic optimal policy at the expense of further exploration. In some environments, narrowing the action distribution can indeed help promote consistent optimal performance. In other environments, this effect can hinder the QD algorithm's ability to branch policies into new cells, given that the outer QD optimization loop relies on gradient estimates produced by PPO to discover unexplored regions of the archive. In environments where we observe this negative effect, we disable gradient flow to the action standard deviation parameter.

Finally, in order to address environmental uncertainty, we insert new policies based on their performance and behavior *averaged* over 10 parallel environments. We leverage GPU acceleration to quickly batch process many parallel environments over a population of branched policies.

### 3.3 CONNECTION TO NATURAL EVOLUTION STRATEGIES

We replace CMA-ES with a Natural Evolution Strategy (NES) to increase the stability and performance of CMA-MAEGA on noisy RL environments. CMA-based variants of PPGA diverged during training. Prior work (Müller & Glasmachers, 2018) showed that CMA-ES struggled to evolve deep neural network controllers with dimensionality $\mathbb{R}^d$ on stochastic RL environments. However, CMA-MAEGA uses CMA-ES to maintain search distribution in objective-measure gradient coefficient space $\mathbb{R}^{k+1} << \mathbb{R}^d$, where $k+1$ can be as small as three dimensions, implying that CMA-ES should still be effective in this low-dimensional space. It was then puzzling to find consistent divergence during the training of our CMA-based algorithm. We hypothesize that the culprit is the cumulative step-size adaptation (CSA) mechanism employed by CMA-ES. CMA-ES uses evolution paths $\rho_\sigma^{(g)}$ to adapt the step size $\sigma^{(g)}$ between successive generations $(g)$. The mechanisms by which $\sigma^{(g)}$ are updated assume a fairly non-noisy and stationary objective $f$. However, the application of CMA-ES to QD optimization on stochastic RL environments presumes the exact opposite. That is, the RL objective $f_{RL}$ is very noisy, and the QD-objective $f_{QD} = g(f_{RL}(\cdot))$, which is a function of the RL objective, is highly non-stationary, since the state of the archive $\mathcal{A}$ *changes* the direction of greatest archive improvement on every iteration. To address the training divergence, we propose using exponential evolution strategies (xNES) Glasmachers et al. (2010), a more recent and theoretically well-motivated method, as a drop in replacement for CMA-ES. Prior works have shown strong links between xNES and CMA-ES, and generalize both methods as instances of *natural evolution strategies* (Akimoto et al., 2010; Glasmachers et al., 2010). In fact, the update step in xNES is equivalent to CMA-ES up to the use of evolution paths. We refer to these prior works for an in-depth comparison. More generally, we believe *any* natural evolution strategy can be used to maintain and update the search distribution over gradient coefficients in this and any prior CMA-based DQD method.

### 3.4 WALKING THE SEARCH POLICY

In standard DQD, $\nabla_{step}$ is computed via weighted linear recombination to produce a gradient vector that steps the search policy in the least explored direction of the archive. However, the resulting gradient vector is a linearized approximation around the current search policy $\theta_\mu$ and thus cannot be reused to take multiple gradient steps in a non-convex optimization problem. It would be remiss not to leverage the highly-parallelized VPPO implementation to "walk" the search policy over *many* steps in the direction of greatest archive improvement. We make the key observation that the mean gradient coefficient vector $\mathbf{c}_\mu$ of the *updated* search distribution maintained by xNES points in the direction of greatest archive improvement for the next iteration of the QD algorithm. Thus, we construct a new multi-objective reward function for VPPO to optimize by taking the dot product between the gradient

coefficient vector and the objective and measure proxies $< c_{\mu_0}, ..., c_{\mu_{k+1}} > \cdot < f, \delta_1, ..., \delta_k >$. Optimizing this function with VPPO allows us to walk the search policy $\theta_\mu$ in the direction of greatest archive improvement by iteratively taking conservative steps, where the magnitude of the movement is controllable by hyperparameter $N_2$. This objective is stationary for all $N_2$ steps, and is only updated after the subsequent QD iteration. We provide pseudocode in Appendix A.

## 4 EXPERIMENTS

We evaluate our algorithm on four different continuous-control locomotion tasks derived from the original Mujoco environments (Todorov et al., 2012): Ant, Walker2d, Half-Cheetah, and Humanoid. The standard objective in each task is to maximize forward progress and robot stability while minimizing energy consumption. We use the Brax simulator to leverage GPU acceleration and massive parallelization of the environments. The observation space sizes for these environments are 87, 17, 18, and 227, respectively, and the action space sizes are 8, 6, 6, and 17, respectively. The standard Brax environments are augmented with wrappers that determine the measures of an agent in any given rollout as implemented in QDax (Lim et al., 2022), where the number of measures of an agent is equivalent to the number of legs. The measure function is the number of times a leg contacts the ground divided by the length of the trajectory. We implement PPGA in pyribs (Tjanaka et al., 2023), with our VPPO implementation based on CleanRL's implementation of PPO (Huang et al., 2022). Most experiments were run on a SLURM cluster where each job had access to an NVIDIA RTX 2080Ti GPUs, 4 cores from a Intel(R) Xeon(R) Gold 6154 3.00GHz CPU, and 108GB of RAM. Some additional experiments and ablations were run on local workstations with access to an NVIDIA RTX 3090, AMD Ryzen 7900x 12 core CPU, and 64GB of RAM.

### 4.1 COMPARISONS

We compare our results to current state-of-the-art QD-RL algorithms: Policy Gradient Assisted MAP-Elites (PGA-ME),[1] Quality Diversity Policy Gradient (QDPG) implemented in QDax (Lim et al., 2022), and CMA-MAEGA(TD3, ES) implemented in pyribs (Tjanaka et al., 2023). We also compare against the state-of-the-art ES-based QD-RL algorithm, separable CMA-MAE (sep-CMA-MAE) (Tjanaka et al., 2022a), which allows evolutionary QD techniques to scale up to larger neural networks. Finally, in order to verify our hypothesis on the emergent synergy between PPO and DQD, we provide an ablation where TD3 is used as a drop-in replacement for PPO in PPGA, which we will refer to as TD3GA going forward. Details on the TD3GA design choices and additional ablations, such as comparing against standard PPO, can be found in the appendix. The same archive resolutions and network architectures are used for all baselines. A full list of shared hyperparameters is in Appendix B. We use an archive learning rate of 0.1, 0.15, 0.1, and 1.0 on Humanoid, Walker2d, Ant, and Half-Cheetah, respectively. Adaptive standard deviation is enabled for Ant and Humanoid. We reset the action distribution standard deviation to 1.0 on each iteration in all other environments.

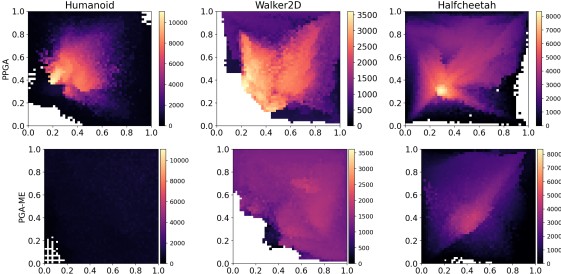

Figure 3: 2D Archive visualizations of PPGA compared to the current state-of-the-art QD-RL algorithm PGA-ME. We use 50x50 archives to show detail.

---

[1]A comparison on Humanoid to PBT-ME (SAC), a recent QD-RL method, can be found in Appendix H. PBT-ME (SAC) was trained with Google TPUs. Due to computational constraints, we were only able to provide a comparison on one task.

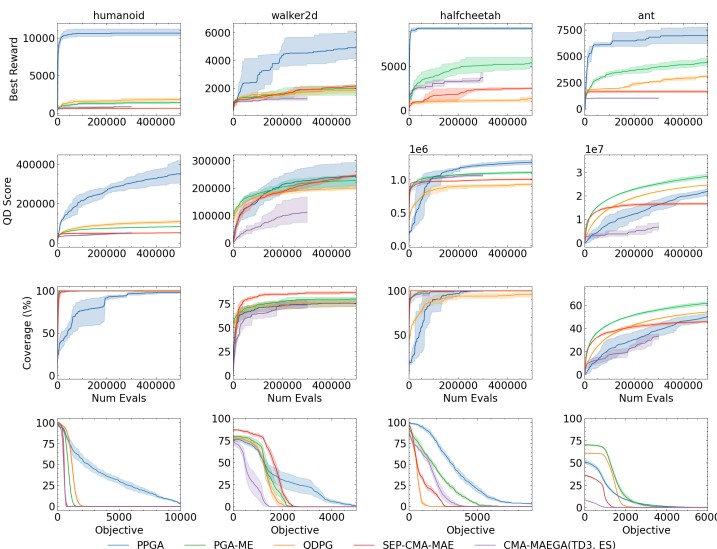

Figure 4: QD metrics and cumulative distributions for archives of PPGA and the baselines. The CCDF plots in the last row indicate the percentage of archive policies above a certain objective threshold. All plots show the mean over four seeds with a 95% bootstrapped confidence interval.

We conduct our experiments using the following criteria: **QD-score**, which is the sum of scores of all nonempty cells in the archive, and **coverage**, which is the percentage of nonempty cells in the archive, have been historically used by QD algorithms to measure performance and diversity respectively, and so we include them as metrics. However, these metrics have a number of edge cases that make them imperfect measures of performance and diversity. For example, an algorithm that fills 100% of the archive with low-performing policies can have a higher QD-score and coverage than a QD algorithm that fills fewer cells with high-performing policies. To more accurately represent the performance and diversity of a given algorithm, we additionally include plots of the **Complementary Cumulative Distribution Function** (CCDF), originally presented in (Vassiliades et al., 2016), which shows what percentage of policies in the archive achieve a reward of $R$ or greater for all possible values of $R$ on the $x$-axis. The CCDF attempts to capture notions of quality of policies in the archive and diversity, while also shedding light on how the policies are *distributed* w.r.t. performance. Finally, we include the **best reward** metric, denoting the highest-performing policy the algorithm was able to discover.

Figures 3 and 4 show that PPGA outperforms baselines in best reward and QD-score, achieving comparable coverage scores on all tasks except Ant, and generating much more illuminated archive heatmaps with a diverse range of higher performing policies than the current state of the art, PGA-ME. Notably, PPGA is the only algorithm that solves Humanoid, achieving over 4x improvement in best-performing policy and QD score compared to baselines. More important than QD-Score and Coverage are the CCDF plots. At $x = 0$, all policies in the archive are included, i.e., $x = 0$ encapsulates the coverage score. CCDF plots provide a better representation of "quality" than QD-score, since we can see how the policies in the archive are distributed. Except for Ant, PPGA produces distributions where more of the mass is distributed to the right where the high-performing policies lie.

In Figure 5, we find evidence that PPO has an important synergy with DQD that is perhaps missing in other RL algorithms. TD3GA fails to find high performing policies on Humanoid. Achieving 100% coverage is indicative of the step size $\sigma$ in xNES exploding and producing highly stochastic policies that, by chance, land in far away cells. This typically occurs when xNES cannot fit a covariance matrix to the data, which in this case are weighted linear combinations of $\nabla f, \nabla \mathbf{m}$ produced by TD3.

## 4.2 POST-HOC ARCHIVE ANALYSIS

QD algorithms are known to struggle with reproducing performance and behavior in stochastic environments. To determine the replicability of our agents, we follow the guidelines in Flageat et al.

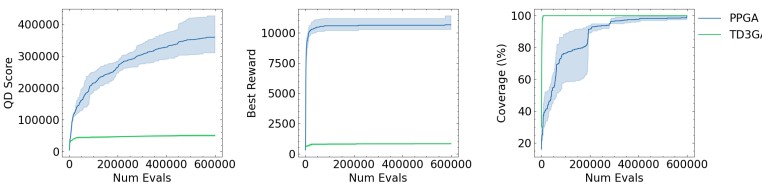

Figure 5: PPGA vs TD3GA on Humanoid on the standard QD metrics. All plots are averaged over 4 seeds. The shaded regions are the 95% bootstrapped confidence intervals.

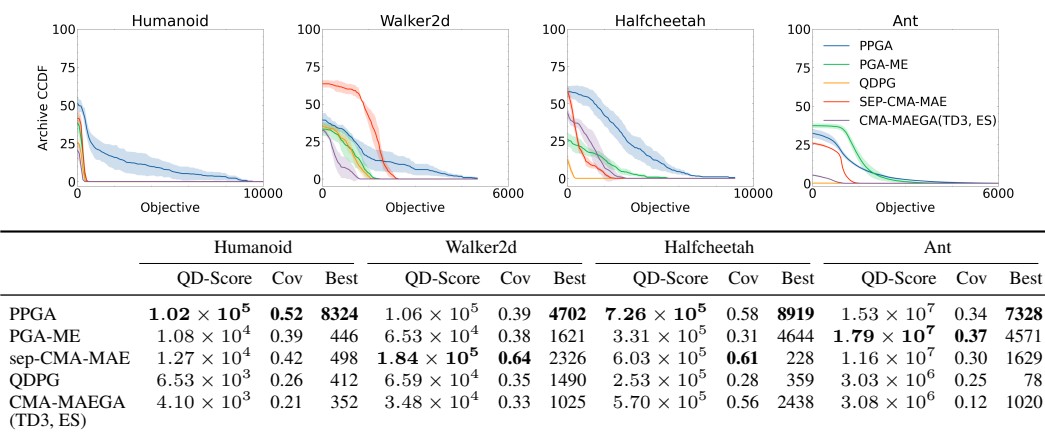

| | Humanoid | | | Walker2d | | | Halfcheetah | | | Ant | | |
|---|---|---|---|---|---|---|---|---|---|---|---|---|
| | QD-Score | Cov | Best | QD-Score | Cov | Best | QD-Score | Cov | Best | QD-Score | Cov | Best |
| PPGA | $\mathbf{1.02 \times 10^5}$ | **0.52** | **8324** | $1.06 \times 10^5$ | 0.39 | **4702** | $\mathbf{7.26 \times 10^5}$ | 0.58 | **8919** | $1.53 \times 10^7$ | 0.34 | **7328** |
| PGA-ME | $1.08 \times 10^4$ | 0.39 | 446 | $6.53 \times 10^4$ | 0.38 | 1621 | $3.31 \times 10^5$ | 0.31 | 4644 | $\mathbf{1.79 \times 10^7}$ | **0.37** | 4571 |
| sep-CMA-MAE | $1.27 \times 10^4$ | 0.42 | 498 | $\mathbf{1.84 \times 10^5}$ | **0.64** | 2326 | $6.03 \times 10^5$ | **0.61** | 228 | $1.16 \times 10^7$ | 0.30 | 1629 |
| QDPG | $6.53 \times 10^3$ | 0.26 | 412 | $6.59 \times 10^4$ | 0.35 | 1490 | $2.53 \times 10^5$ | 0.28 | 359 | $3.03 \times 10^6$ | 0.25 | 78 |
| CMA-MAEGA (TD3, ES) | $4.10 \times 10^3$ | 0.21 | 352 | $3.48 \times 10^4$ | 0.33 | 1025 | $5.70 \times 10^5$ | 0.56 | 2438 | $3.08 \times 10^6$ | 0.12 | 1020 |

Figure 6: Corrected CCDFs and Corrected QD metrics: QD-Score, *Cov*erage, *Best* Reward. Results are averaged over four seeds with error bars showing a 95% bootstrapped confidence interval.

(2023). We re-evaluate each agent in the archive 50 times and average its performance and measures to construct a Corrected Archive and use this to produce Corrected QD metrics such as QD-Score and Coverage. Fig. 6 shows the corrected QD metrics and the corrected CCDFs, respectively. After re-evaluation, PPGA maintains the lead in best reward on all tasks, QD-score on Humanoid and Ant, and Coverage on Humanoid. The CCDF plots of the Corrected Archives show PPGA producing better distributions of policies on all tasks but Ant, suggesting PPGA's policies are robust to stochasticity.

## 5 DISCUSSION AND LIMITATIONS

We present a new method, PPGA, which is one of the first QD-RL methods to leverage on-policy RL, the first to solve the challenging Humanoid task, and the first to achieve equivalent performance in best reward compared to standard RL on all domains. We show that DQD algorithms and on-policy RL have emergent synergies that make them work particularly well with each other. However, instead of simply combining DQD and on-policy RL as is, we re-examine the fundamental assumptions and mechanisms of each component and implement changes that maximize their synergies. There are some caveats with this approach. On-policy RL algorithms such as PPO are quite sample-inefficient and require many parallel environments per agent in order to compute the stochastic policy gradient. Although GPU acceleration and massive parallelism improve wall-clock convergence over off-policy RL, this makes our approach less sample-efficient than other off-policy QD-RL methods. Secondly, enabling PPO's adaptive standard deviation parameter (which is true by default for PPO) can have detrimental effects on PPGA's exploration capabilities, as made evident by the coverage score on Ant. This is mainly due to the fact that PPO favors collapsing the standard deviation to achieve higher average returns. In the future, we will investigate modifying the standard deviation parameter such that it dynamically shrinks or increases the standard deviation value based on the QD-optimization landscape as opposed to the RL one. Finally, we are interested to see how this method scales to even more data-rich regimes such as distributed settings, as well as its application to harder problems such as real robotics tasks. We leave these as potential avenues of future research.

## 6 REPRODUCIBILITY

In the supplemental material, we provide the source code and training scripts used to produce our results. In the README, we include documentation for setting up a Conda environment, running our training scripts, and visualizing our results. In addition, we provide pre-trained archives whose results were presented in this work. Detailed pseudocode and a list of relevant hyperparameters can be found in Appendices A and B.

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

## A   PPGA PSEUDOCODE

---

**Algorithm 1** Proximal Policy Gradient Arborescence

---

**Input:** Initial policy $\theta_0$, VPPO instance to approximate $\nabla f, \nabla \mathbf{m}$ and move the search policy, number of QD iterations $N_Q$, number of VPPO iterations to estimate the objective-measure functions and gradients $N_1$, number of VPPO iterations to move the search policy $N_2$, branching population size $\lambda$, and an initial step size for xNES $\sigma_g$

Initialize the search policy $\theta_\mu = \theta_0$. Initialize NES parameters $\mu, \Sigma = \sigma_g I$
**for** iter $\leftarrow 1$ **to N do**
   $f, \nabla f, \mathbf{m}, \nabla \mathbf{m} \leftarrow VPPO.compute\_jacobian(\theta_\mu, f(\cdot), \mathbf{m}(\cdot), N_1)$
   $\nabla f \leftarrow \text{normalize}(\nabla f), \nabla \boldsymbol{m} \leftarrow \text{normalize}(\nabla \boldsymbol{m})$
   $\_ \leftarrow \text{update \_archive}(\theta_\mu, f, \boldsymbol{m})$
   **for** $i \leftarrow 1$ **to** $\lambda$ **do**
      $c \sim \mathcal{N}(\mu, \Sigma)$ // sample gradient coefficients
      $\nabla_i \leftarrow c_0 \nabla f + \sum_{j=1}^{k} c_j \nabla m_j$
      $\theta_i' \leftarrow \theta_\mu + \nabla_i$
      $f', *, m', * \leftarrow \text{rollout}(\theta_i')$
      $\Delta_i \leftarrow \text{update\_archive}(\theta_i', f', \boldsymbol{m'})$
   **end for**
   rank gradient coefficients $\nabla_i$ by archive improvement $\Delta_i$
   Adapt xNES parameters $\mu = \mu', \Sigma = \Sigma'$ based on improvement ranking $\Delta_i$
   $f'(\theta_\mu) = c_{\mu,0} f + \sum_{j=1}^{k} c_{\mu,j} m_j$, where $\mathbf{c}_\mu = \mu'$ // construct multi-objective reward function
   $\theta_\mu' = VPPO.train(\theta_\mu, f', N_2)$ // standard PPO training procedure
   **if** *there is no change in the archive* **then**
      Restart xNES with $\mu = 0, \Sigma = \sigma_g I$
      Set $\theta_\mu$ to a randomly selected existing cell $\theta_i$ from the archive
   **end if**
**end for**

---

**Algorithm 2** Update Archive

---

**Input:** Solution $\theta$ to insert, episodic reward $f$, measures $\mathbf{m} = <m_1, ..., m_k>$, archive $\mathcal{A}$, archive learning rate $\alpha$
$\theta_{inc}, f_{inc} = \mathcal{A}[\mathbf{m}]$ if $\mathcal{A}[\mathbf{m}]$ is nonempty else $None, 0$ // incumbent policy
$\Delta_i = 0$
**if** $f > f_{inc}$ **then**
   insert $\theta$ into cell $\mathcal{A}[\mathbf{m}]$
   $f_{inc} \leftarrow (1 - \alpha) f_{inc} + \alpha f$
   $\Delta_i = f - f_{inc}$
**end if**
return $\Delta_i$

---

---

**Algorithm 3** Vectorized-PPO (VPPO)

---
**Input:** Initial search policy $\pi_{\theta_i}$, objective functions to optimize $\mathbf{f} = f_1(\cdot), ..., f_k(\cdot)$, number of VPPO iterations $N$, number of parallel environments $E$, rollout length $L$

Initialize the vectorized agent $\overrightarrow{\pi}_{\theta_i} = $ vectorized_agent$([\pi_\theta] \times (k+1))$
**for** iter $\leftarrow 1$ **to** $N_1$ **do**
    $(\mathbf{S}, \mathbf{A}, \mathbf{R}, \mathbf{S'}) \leftarrow$ rollout(vectorized_agent, $E$, $L$, $\mathbf{f}$) // Note that $\mathbf{S} = \{S_1, ..., S_k\}$, etc
    advantage $\mathbb{A}$, returns $\mathcal{G} \leftarrow$ batch_calculate_rewards$(\mathbf{S}, \mathbf{A}, \mathbf{R}, \mathbf{S'}, \mathbf{f})$
    $\overrightarrow{\pi}_\theta{}' \leftarrow$ batch_gradient_descent$(\mathbb{A}, \mathcal{G}, \overrightarrow{\pi}_\theta)$ // using the stochastic policy gradient
    $\overrightarrow{\pi}_\theta \leftarrow \overrightarrow{\pi}_\theta{}'$
**end for**
$\nabla\mathbf{f} \leftarrow \overrightarrow{\pi}_\theta{}' - \overrightarrow{\pi}_{\theta_i}$
return $\nabla\mathbf{f}$

---

## B    Hyperparameters

Table 1: List of relevant hyperparameters for PPGA shared across all environments.

| Hyperparameter | Value |
|---|---|
| Actor Network | [128, 128, Action Dim] |
| Critic Network | [256, 256, 1] |
| $N_1$ | 10 |
| $N_2$ | 10 |
| PPO Num Minibatches | 8 |
| PPO Num Epochs | 4 |
| Observation Normalization | True |
| Reward Normalization | True |
| Rollout Length | 128 |

## C    Ablation Against CMA-MAEGA

PPGA makes two key changes compared to standard DQD algorithms such as CMA-MAEGA: Walking the search policy with VPPO vs. weighted linear recombination of the gradients and replacing xNES. We compare walking the search policy with VPPO to using gradient recombination in the original formulation of CMA-MEGA. In addition, we ran an ablation using xNES as the outer-loop optimizer compared to CMA-ES. However, the step-size adaptation parameter quickly diverges with CMA-ES and destabilizes training, and thus were unable to provide plots for this ablation. Fig. 7 shows the comparison of walking the search policy with VPPO versus using weighted linear recombination of the gradients. We believe that the large gap in performance is due to the fact that we can take multiple steps with VPPO via the $N_2$ hyperparameter. Weighted recombination results in a single gradient step where the updated search policy may land too close to the previous search policy's cell. When we branch from the updated search policy, many-branched agents will fall into overlapping cells, resulting in small archive improvement, which can lead to the emitter prematurely leaving a high-performing region of the search space.

## D    TD3GA Implementation Details

As part of our ablation study, we implemented TD3GA, which differs from PPGA by replacing all PPO mechanisms with TD3 (Fujimoto et al., 2018). This implementation required several algorithmic decisions, as PPGA was originally designed to integrate with an on-policy method like PPO rather than an off-policy method like TD3, In this section, we describe these decisions. In general, we intend our decisions to make TD3GA operate as closely as possible to PPGA, and we leave it to future work to explore whether variations of these decisions will further improve performance.

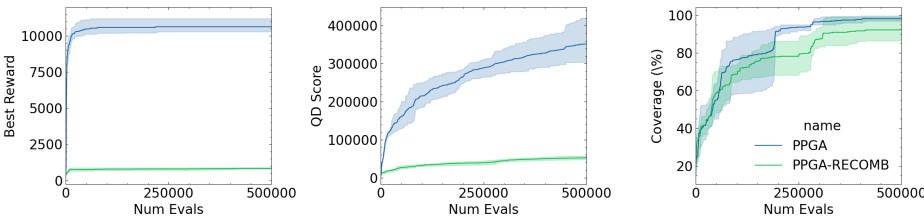

Figure 7: Ablation of walking the search policy with VPPO vs gradient recombination done in CMA-MEGA/CMA-MAEGA on the Humanoid environment.

**Background:** TD3 is an off-policy actor-critic method designed for single-objective RL tasks. TD3 maintains an actor (i.e., a policy) that takes actions in the environment and a critic that estimates the action-value function. TD3 also maintains a replay buffer that stores experiences collected by the actor. Over time, the actor is trained to optimize the critic. Simultaneously, based on experience in the replay buffer, the critic learns to better predict the action-value function.

**Design Decisions:**

1. **Number of critics:** In TD3GA, we maintain a TD3 critic for the objective function and one for each of the measure functions. We also create a separate critic for the weighted objective that is used when walking the search policy. We refer to these critics as the *objective* critic, *measure* critics, and *walking* critic.

2. **Choice of actor for critic training:** When training the critic, TD3 requires an actor that generates actions for states sampled from the replay buffer. Prior QD-RL methods that estimate objective gradients with TD3, e.g., PGA-ME (Nilsson & Cully, 2021) and CMA-MEGA (TD3, ES) (Tjanaka et al., 2022b), fulfill this role with a dedicated actor. This actor is referred to as a *greedy actor* since its primary purpose is to optimize its performance with respect to the critic.

   In theory, since TD3 is an off-policy method, any actor, including a greedy actor, can be used to train the critic. However, to make TD3GA closer to PPGA, we instead use a copy of the current search policy to train the critic. This decision provides our TD3 instances with an on-policy nature that mirrors the PPO mechanisms found in PPGA.

3. **When to train critics:** Similar to the PPO value functions in PPGA, we maintain all critics throughout the entire training run, updating them on every iteration.

4. **Experience collection:** This decision concerns which experience collected in the environment should be inserted into the replay buffer. With the settings in our paper, PPGA (and TD3GA) samples 300 policies per iteration, and each policy is evaluated for 10 episodes, with each episode having up to 1,000 timesteps of experience; in total, these policies generate 3 million timesteps per iteration. The typical replay buffer size (Fujimoto et al., 2018) in TD3 is 1 million, meaning the buffer would be filled three times over if we inserted all of this experience, i.e., the critic would never be trained with the experience from two-thirds of all policies.

   To ensure that we collect experience from many policies across many iterations, we make two decisions. First, we only collect one episode of experience from each agent — this already cuts down experience collected on each iteration to 300,000 timesteps. Second, we increase the replay buffer size to 5 million to store experience across more iterations.

   Note that there is no analog to the replay buffer in PPGA since PPO is an on-policy method. Instead, PPO regresses the value function based on experience collected while evaluating the policy.

5. **Gradient computation:** We begin by describing how a prior method estimates the objective gradient with TD3. To estimate the objective gradient of a policy, CMA-MEGA (TD3, ES) samples a batch of experience and passes the batch through the corresponding actor. The actions outputted by the actor are then inputted to the critic. Backgpropagating through the critic and the actor then provides the objective gradient.

Roughly, the above procedure corresponds to taking a *single* step of TD3. However, in PPGA, the gradient is computed by taking the difference after *multiple* steps of PPO (the number of steps is determined by the hyperparameters $N_1$ and $N_2$. The straightforward approach to mirror this behavior in TD3GA is to also output a gradient after several steps of TD3. Thus, when training each critic, we also track the final state of the actor (recall that the actor used in critic training is a copy of the search policy). At the end of critic training, our gradient is then the difference between the final actor and the original search policy.

For example, to compute the objective gradient, we train the objective critic with an actor that is a copy of the search policy. While training the critic, the actor is updated so that it maximizes the objective critic. After $N_1$ steps, we compute the objective gradient as the difference between the actor and the original search policy.

6. **Actor target networks:** One TD3 mechanism that improved stability and performance was target networks, which are slowly updating versions of the actor and critic parameters. In TD3GA, it is straightforward to apply this mechanism to the critic parameters.

   However, since the actor is reset to the current search policy on every iteration, it is difficult to maintain a single target network. Thus, on every iteration, the target network for the actor is simply reset to the current search policy's parameters before the gradient computation.

**Pseudocode:** Algorithm 4 shows the process for computing an objective gradient in TD3GA. The same process applies to computing measure gradients. The process for walking the search policy is also similar, except that the reward is a weighted combination of the objective and measures (the weights come from the mean $\mu$ of the emitter's coefficient distribution). Furthermore, when walking the search policy, we return the final actor instead of a gradient.

**Hyperparameters:** TD3GA inherits all relevant hyperparameters from PPGA (Table 1). It also inherits TD3 hyperparameters from prior QD-RL works that incorporate TD3 (Tjanaka et al., 2022b; Nilsson & Cully, 2021), except for having a larger replay buffer. We also increase the batch size used during critic training to mimic the batch size used by PPO to regress the value function. Table 2 lists hyperparameters shared across all environments. Note that the archive learning rate depends on the environment but is identical to that used in PPGA.

Table 2: List of hyperparameters used in TD3GA, shared across all environments.

| HYPERPARAMETER | VALUE |
| --- | --- |
| ACTOR NETWORK | [128, 128, ACTION DIM] |
| CRITIC NETWORK | [256, 256, 1] |
| TRAINING STEPS FOR OBJECTIVE AND MEASURES ($N_1$) | 10 |
| TRAINING STEPS FOR WALKING ($N_2$) | 10 |
| OPTIMIZER | ADAM |
| ADAM LEARNING RATE | $3 \times 10^{-4}$ |
| TARGET NETWORK UPDATE RATE ($\tau$) | 0.005 |
| TARGET NETWORK UPDATE FREQUENCY ($d$) | 2 |
| SMOOTHING NOISE STANDARD DEVIATION ($\sigma_p$) | 0.2 |
| SMOOTHING NOISE CLIP ($c$) | 0.5 |
| DISCOUNT FACTOR ($\gamma$) | 0.99 |
| REPLAY BUFFER SIZE | 5,000,000 |
| BATCH SIZE ($n_{batch}$) | 48,000 |

**Algorithm 4** TD3 Gradient Computation for the Objective. Adapted from TD3 (Fujimoto et al., 2018)

---

**Input:** Current search policy params $\theta_\mu$, current TD3 critic networks $Q_{\psi_1}$ and $Q_{\psi_2}$ parameterized by $\psi_1$ and $\psi_2$ respectively, current critic targets $\psi_1'$ and $\psi_2'$, replay buffer $B$

**Hyperparameters:** Training steps $N_1$ or $N_2$, target network update rate $\tau$, target network update frequency $d$, smoothing noise standard deviation $\sigma_p$, smoothing noise clip $c$, discount factor $\gamma$, batch size $n_{batch}$

Initialize actor $\phi = \theta_\mu$, actor target $\phi' = \theta_\mu$

{*Either $N_1$ for gradient computation or $N_2$ for walking the search policy*}
**for** iter $\leftarrow 1$ **to** $N_1$ **do**
    {*$r$ is replaced with the measures for measure gradients, or a weighted combination of the reward and measures for walking the search policy*}
    Sample mini-batch of $n_{batch}$ transitions $(s, a, r, s')$ from $B$

    {*Train the critics*}
    $\tilde{a} \leftarrow \pi_{\phi'}(s') + \epsilon, \epsilon \sim \text{clip}(\mathcal{N}(0, \sigma_p), -c, c)$
    $y \leftarrow r + \gamma \min_{i=1,2} Q_{\psi_i'}(s', \tilde{a})$
    {*This update is performed with Adam*}
    Update critics $\psi_i \leftarrow \arg\min_{\psi_i} \frac{1}{N} \sum (y - Q_{\psi_i}(s, a))^2$

    **if** $t \mod d$ **then**
        {*Update $\phi$ by the deterministic policy gradient with Adam*}
        $\nabla_\phi J(\phi) = \frac{1}{n_{batch}} \sum \nabla_a Q_{\psi_1}(s, a)|_{a=\pi_\phi(s)} \nabla_\phi \pi_\phi(s)$
        {*Update target networks:*}
        $\psi_i' \leftarrow \tau\psi_i + (1 - \tau)\psi_i'$
        $\phi' \leftarrow \tau\phi + (1 - \tau)\phi'$
    **end if**
**end for**

{*Note that $\psi_1$, $\psi_2$, $\psi_1'$, and $\psi_2'$ are maintained across calls to this function*}

{*When walking the search policy, we just return the new policy params $\phi$*}
$\nabla f \leftarrow \phi - \theta_\mu$
**return** $\nabla f$

---

## E  HYPERPARAMETER STUDY

We investigate the effects of changing the $N_1$ and $N_2$ hyperparameters on the performance of PPGA in the Humanoid domain. We run 4 seeds of PPGA on the following combinations of $N_1$ and $N_2$: (10, 5), (5, 10), and (1, 1), and compare against the baseline (10, 10). With these experiments, we wish to address the following questions

1. How few PPO steps on both the objective-measure Jacobian calculation and walking the current search policy can we take before noticing performance degradation?

2. How do asymmetric choices for $N_1$ and $N_2$ (i.e. more Jacobian calculation steps than walking steps and vice versa) affect performance?

(5, 10) achieves the most consistent results while achieving the same performance as the baseline, while (10, 5) results in very high run-to-run variance and lower coverage. This suggests that spending more computation on walking the current search policy is more important than approximating the objective-measure Jacobian. Nonetheless, we conclude that in a more computationally constrained setting, PPGA could be tuned to perform fewer $N_1$ and $N_2$ steps while still maintaining good performance. Finally, (1, 1) performs the worst, achieving slightly more than 50% of the baseline's performance, implying that there is significant performance degradation when too few Jacobian-calculation and walking steps are taken.

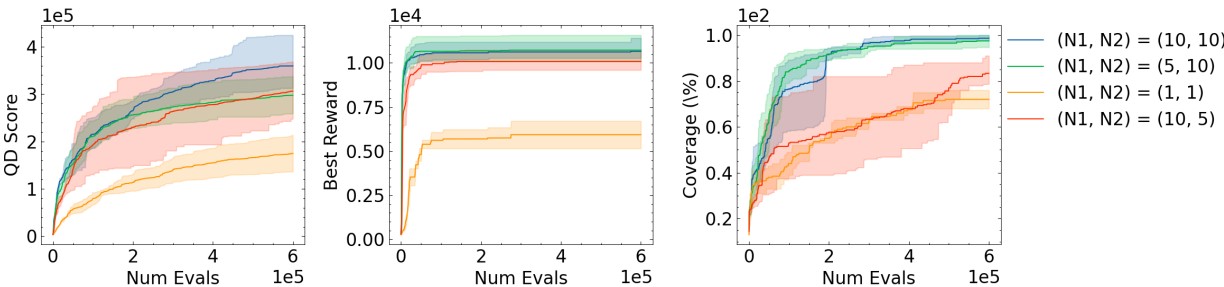

Figure 8: Study of the effect of different hyperparameter choices for $N_1$ and $N_2$. $N_1$ is the number of PPO steps used to calculate the objective-measure Jacobian, and $N_2$ is the number of PPO steps used to walk the search policy. All plots are averaged over 4 seeds. The shaded region is the 95% boostrapped confidence interval.

## F  PPO ABLATIONS

We perform ablations against standard PPO on Humanoid, the most challenging of all domains, in order to compare PPGA's relative performance against standard RL. We first ablate how the measure functions affect performance. The algorithm "PPGA (No Measure)" performs exactly as standard PPGA but with no measure functions. In this case, PPGA computes gradients for the RL objective when branching and deciding where in the archive to move next. Unsurprisingly, PPGA (No Measures) achieves the same best reward as PPGA, but with less than half the archive coverage.

In the second ablation, we run vanilla PPO and store the intermediate policies in between mini-batch gradient descent steps in an archive. The algorithm, dubbed "PPO + Archive", achieves the same best reward as PPGA, but with less than 1% archive coverage. Note that PPGA (No Measures) is still performing an outer-loop optimization step of the QD-objective, thus achieving better coverage than PPO + Archive, whereas PPO + Archive only optimizes for the RL objective.

| Algorithm | QD-Score | Coverage | Best Reward |
|---|---|---|---|
| PPGA | $3.59 \times 10^5$ | 98.67% | 9677 |
| PPGA (No Measures) | $8.24 \times 10^4$ | 32.06% | 9653 |
| PPO + Archive | $3.30 \times 10^3$ | 0.08% | 9651 |

Table 3: Ablation study of PPGA with various components on Humanoid. PPGA (No Measures) functions exactly as PPGA, but only computes objective gradients and walks the search policy with respect to $f$. PPO + Archive runs standard PPO and stores the intermediate policy updates as policies into an archive. Archives are 10x10.

## G    SCALING EXPERIMENTS

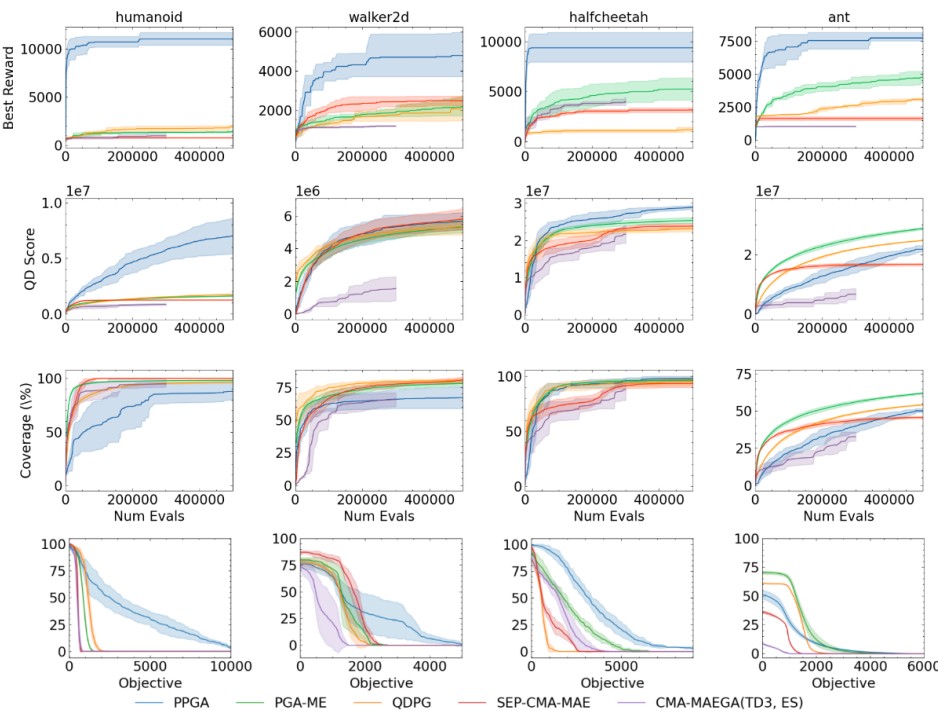

Figure 9: QD-Score, Coverage, Best Reward, and CCDF plots for PPGA and baselines, with 50x50 archives for all tasks except for Ant. Ant retains the same $10^4$ archive resolution, as this is already sufficiently large.

We test the scalability of PPGA along two axes – the ability to scale to larger archives and the ability to learn with more data. To test for the first property, we scale up the archive resolution to 50x50 for locomotion tasks with two measures i.e. Humanoid, Walker2D, and Halfcheetah and compare against baselines. PPGA retains the same performance across tasks, with slight reductions in coverage. This is due to PPGA being an entirely gradient-based method, using gradients for branching and walking the search policy, whereas prior methods that employ ES can get lucky and randomly mutate policies into far away cells.

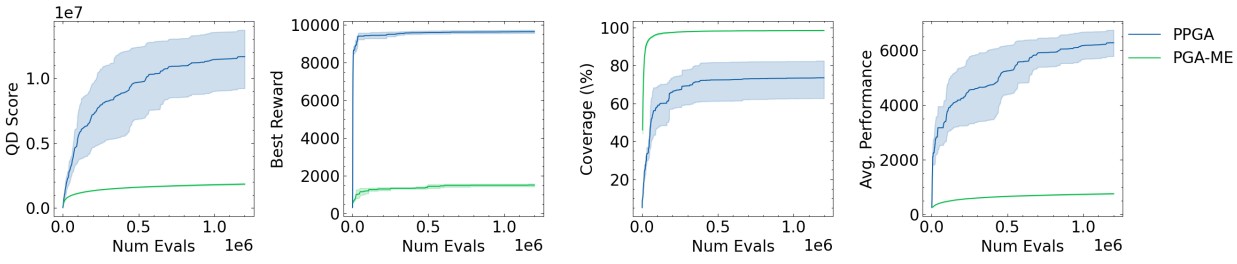

Figure 10: QD Metrics for 50x50 archives of PPGA and PGA-ME trained to 1.2 million evaluations. Results are averaged over 4 seeds. The additional "average performance" metric is presented to show how performance over all policies in the archive changes with additional training.

Finally, we run our algorithm on a 50x50 archive with 1.2 million evaluations, more than twice as long as the main experiments, of both PPGA and the state of the art baseline PGA-ME, on Humanoid. We report an additional metric, the average performance of the archive, which is the sum of scores of all policies divided by the total number of policies in the archive. This can also be interpreted as the normalized QD score. We find that PPGA continues to improve on this metric, indicating that the episodic returns and thus performance of many policies in the archive continues to improve with additional training. We observe this effect to a much lesser extent with PGA-ME.

## H COMPARING TO PBT-ME (SAC)

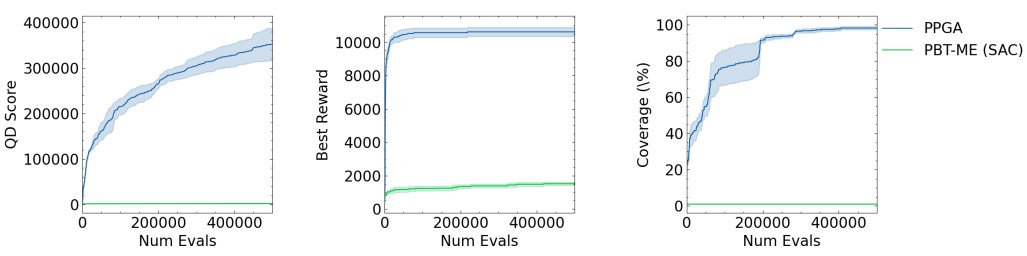

Figure 11: Comparison to PBT-ME (SAC) on Humanoid, which uses Soft Actor-Critic (Haarnoja et al., 2018) to compute the policy gradient when improving the agents.

Population-Based Training MAP-Elites (PBT-ME) (Pierrot & Flajolet, 2023) is a recent QD-RL algorithm that alleviates hyperparameter sensitivity in QD-RL algorithms by evolving populations of agents and their hyperparameters, while also using the policy gradient formulation to improve the agents' performance. The evolved and optimized policies are added to an archive following the MAP-Elites formulation. PBT-ME was run with Google TPUs – due to computational constraints, we were only able to make comparisons against PBT-ME on the Humanoid domain, which we present here. Specifically, we compare to the PBT-ME (SAC) variant.

