# OpenReview forum: "Proximal Policy Gradient Arborescence for Quality Diversity Reinforcement Learning"
_ICLR.cc/2024/Conference — ICLR 2024 spotlight_

### Official Review · Reviewer_vTCA · 2023-10-24

**Soundness:** 3 good
**Presentation:** 3 good
**Contribution:** 4 excellent
**Rating:** 8
**Confidence:** 5

**Summary:**

The authors propose PPGA, a QDRL algorithm where the RL part is based on PPO, an on-policy RL algorithm, whereas all existing QDRL algorithms use the off-policy PPO. This change makes it possible to benefit from the good parallelization capabilities of PPO, and the authors claim that there are also other interesting synergies between PPO and their QD approach. The authors show that PPGA outperforms previous SOTA QDRL algorithms in four locomotion benchmarks.

After the authors rebuttal and given their revised version, I change my evaluation to "accept".

**Strengths:**

- The proposed PPGA algorithm is novel and interesting. It outperforms relevant previous SOTA baselines. To me, this should be sufficient to warrant publication.

- The experimental study looks correct.

**Weaknesses:**

- The paper (still) needs improvements in the presentation to make it more interesting to a large machine learning audience.

- The claim about the synergies between DQD and PPO looks insufficiently backed-up. In particular, the main paper does not even mention the TD3GA algorithm, while the study of combining DQD with TD3 is crucial to understand these synergies. More generally, your central claim is that using on-policy RL better fits the DQD framework, so the comparison to TD3GA should be central.

**Questions:**

To be fully transparent, this is the third time I'm reviewing this paper, after ICML23 and NeurIPS23. Each time I was in favor of accepting the paper because I believe the proposed algorithm is truly interesting, but each time the authors failed to convince some reviewers, mostly due to some writing issues and insufficient focus on the comparison between TD3GA and PPGA. I hope this time could be the right time with a further writing effort that I will try to contribute to below. So my review will mostly focus on writing aspects, but keep in mind that the TD3GA vs PPGA comparison is crucial and not even mentioned in the main paper.

Abstract:

- "Existing QD-RL approaches have been constrained to sample efficient, deterministic off-policy RL algorithms and/or evolution strategies, *and struggle with highly stochastic environments*." -> the rest of the abstract makes it clear that you switch to on-policy RL, but it does not make clear that you will solve the highly stochastic environments issue. Do you really need to mention that? If yes, you need to mention how you will solve it.

- "propose several changes" -> changes wrt what? It is a weird formulation, you have a new algorithm, you propose new things, no need to mention that...

Introduction:

parag 1 is OK (maybe a ref at the end of the sentence finishing with "high-quality solutions" or merge the sentence with the next ones, where the references are).

parag 2 has several issues:

- "For example, all QD-RL algorithms for locomotion to date use non-differentiable measure functions, which in most cases prevents direct RL optimization."

-> At this point, the standard ML reader does not know what a measure function is,

-> I don't agree that having non-differentiable measure functions prevents direct RL optimization. RL methods are precisely good at approximating some gradients when they are not available. Even if you want to discuss this with me, this sentence is ill-placed, as the standard reader cannot get its meaning.

- "Prior methods that investigated combining DQD". You should re-explain the acronym, doing it once in the abstract is not enough.

- "However, given the gap in performance between standard RL and QD-RL algorithms in terms of best-performing policy, we believe that DQD algorithms, under a different formulation more synergistic with its underlying mechanisms, can close this gap. We instead believe that one or both of existing DQD and RL methods must be adapted to each other in order to maximally exploit their synergy."

-> Both sentences say more or less the same thing, I suspect this is a last-minute-no-reread edition issue.

- "maintain a single search point within the archive that" -> at this point, the standard ML reader does not know what the search point is, and what the archive is.

- "The question we aim to answer becomes clear through this high-level view – can we better adapt DQD algorithms to challenging RL domains by exploiting its synergy with PPO?" -> the third time you mention exploiting this synergy in the same paragraph...

I afraid that, with all the points above, you have already lost the standard reader, who won't read your paper any further. Instead, you must give a high-level view of PPGA from the RL viewpoint, with the RL vocabulary (a search point is a policy, etc.).

- As contribution (1), you propose an implementation ... : I see no link to a source code in the paper, so you should not mention such a contribution [EDIT: OK, the source code is in the supplementary, forget this comment]

- What a "non-stationary QDRL task"?

- Contrib (4) sould be reformulated with the RL vocabulary. Otherwise, rather submit your paper to GECCO, as I already advised you twice...

- where T is (the) episode length

- the deep neural network represents *a* state-action mapping, not necessarily the optimal one.

### Deep RL background:

- About your Deep RL background, it is close to good (there is no major mistake), but it is still confusing for the non-expert and you can do a better job.
- I think the point you want to make is that PPO is on-policy. In the current classification, it is not in the "on-policy" part, but in the "trust region" part
- Mentioning actor-critic in the on-policy part the way you do brings confusion. It is true that A2C and A3C are actor-critic and on-policy, but DDPG, TD3, SAC and others are actor-critic too and they are off-policy.
- I think what you need is the distinction between policy-based methods (Reinforce, A2C, TRPO, PPO) which derive their gradient from the policy and are on-policy, and value-based methods (DQN, DDPG, TD3, SAC, ...) which derive their gradient from the critic and are off-policy.

See e.g. Nachum, O., Norouzi, M., Xu, K., & Schuurmans, D. (2017). Bridging the gap between value and policy based reinforcement learning. Advances in neural information processing systems, 30.

### Other points

- the equation with L(theta) should finish with a dot.

- In the QD optimization section, don't you want to mention QDRL in the last parag?

- "gradients around the search point ∇ 1 , ..., ∇ λ" -> gradients ∇ 1 , ..., ∇ λ around the search point.

- Again, why not call the search point "the current policy"? The same for "solutions", they are policies.

- "some minimum threshold". Why call this a minimum threshold? Threshold for what? I rather see it as an exploration bonus...

- "in the direction of the archive that is least" -> doesn't this also take performance into account?

- In 2.4: "In prior work (ref) ... In this work" -> You should not write this in a way that let us know who you are. *Your paper could be desk-rejected for that. Actually, this is transparent given the (too) many self-references, but you have to follow the rules*.

- In 3.2, I'm afraid the claim that trust region methods provide formal guarantees on the quality of the gradient estimates is wrong. The proof in TRPO comes with unrealistic assumptions that are always violated in experimental RL work.

- You use the MMPs as rewards to optimize. So it means that your algorithm is looking for as much leg contact as possible. Shouldn't it be looking for as much "leg contact diversity" as possible instead? This is unclear. Could you clarify?

- "We additionally modify the computation of the policy gradient into a batched policy gradient method, where intermediate gradient estimates of each function w.r.t. policy params only flow back to the parameters corresponding to respective individual policies during minibatch gradient descent." -> This part is very unclear to me, a small diagram or equations would probably help.

- jacobian -> Jacobian

- There are many considerations about using constant variance or not. Actually in the literature, there are 3 options: using a constant variance, using a tunable variance which is not a function of states, and using a tunable state-dependent variance where the NN outputs a mean and a variance for each state. The second one is used e.g. in the TRPO paper, see Fig. 3 here: https://proceedings.mlr.press/v37/schulman15.pdf
I think you need to further study this point, your work is not convincing in that respect. The fact that you "disable gradient flow" to the variance parameter or not depending on the environment is not satisfactory at all.

- About Section 3.3, I think the first paragraph which establishes that xNES is a better option that CMA-ES in your context could be moved in an appendix together with the corresponding sutdy, as this is not central to your story. You could make profit of the earned space to add a paragraph about the comparison between TD3GA and PPGA (with selected results), which is much more important.

- At the end of 3.4, you mention the outer optimization loop, but the inner/outer loop distinction has not been made explicit in the main paper.

- "We use an archive learning rate of 0.1, 0.15, 0.1, and 1.0..." -> The fact that there is such a learning rate is not explained before.

- All experimental figures and tables make a poor use of space. By reworking this aspect you can both save more space and make your results more readable and explicit.

- In Fig. 4, using +/- one std as variance information is a bad practice in RL, where the variance is generally not Gaussian. See
Patterson, A., Neumann, S., White, M., & White, A. (2020). Draft: Empirical Design in Reinforcement Learning. Journal of Artificial Intelligence Research, 1. and Agarwal, R., Schwarzer, M., Castro, P. S., Courville, A. C., & Bellemare, M. (2021). Deep reinforcement learning at the edge of the statistical precipice. Advances in neural information processing systems, 34, 29304-29320, the latter comes with a useful visualization library.

- Figure 4 is cited before fig 3, you should reorder

- "We present a new method, PPGA, which is one of the first QD-RL methods to leverage on-policy RL," -> one of the first, so what are the others?

- "We show that DQD algorithms and on-policy RL have emergent synergies that make them work particularly well with each other." -> I'm sorry but this point does not emerge clearly from reading the paper. You should have somewhere a paragraph about the investigations of these synergies.

- Your paper needs slightly more than 9 pages. Again, *it could have been desk-rejected for that*. Please follow the rules.

---

> ### Author Response · Authors · 2023-11-16
> **Response to Reviewer vTCA Part 1**
>
> We appreciate the incredibly in-depth feedback. Points on writing brought up by the reviewer that require no discussion (including the reviewer's main concern on making the TD3GA comparison central) have been directly implemented in the paper, and we have updated the pdf accordingly. For points that require discussion, we respond to them individually:
>
> **the rest of the abstract makes it clear that you switch to on-policy RL, but it does not make clear that you will solve the highly stochastic environments issue.**
>
> PPGA inserts policies into the archive based on their average performance and behavior over 10 parallel environments. We find empirical evidence that this indeed improves robustness to uncertainty. We have added this point to the end of section 3.2. Section 4.2 also investigates PPGA's robustness to uncertainty.
>
> **"propose several changes" -> changes wrt what?**
>
> We wish to convey other changes, such as replacing CMA-ES with xNES, etc., that are subtle improvements over existing work that have large implications for our method (ex. CMA-ES destabilizing training). To improve clarity, we changed this to read "propose additional improvements over prior work".
>
> **At this point, the standard ML reader does not know what a measure function is...**
>
> **I don't agree that having non-differentiable measure functions prevents direct RL optimization...**
>
> Per the reviewer's suggestions, we have changed this to read "For example, all QD-RL algorithms for locomotion to date use non-markovian measures of behavioral diversity, which in many cases prevents direct RL-optimization."
>
> **the third time you mention exploiting this synergy in the same paragraph...**
>
> We changed this to read "It is through this high level view that we see the emergent synergy between PPO and DQD, in that PPO can be used to collect gradient estimates from online data when one or both of the objective and measure functions are markovian and non-differentiable."  We have also removed redundant uses of the word "synergy" from the manuscript.
>
> **What a "non-stationary QDRL task"**
>
> We have expanded on this definition at the beginning of section 3.2.
>
> **I think the point you want to make is that PPO is on-policy. In the current classification, it is not in the "on-policy" part, but in the "trust region" part**
>
> We wish to make both points i.e. the on-policy classification is synergistic with DQD, and that the approximate trust region is a nice property to have when the reward function is non-stationary.
>
> **Mentioning actor-critic in the on-policy part the way you do brings confusion...**
>
> We have updated the DeepRL background section according to the reviewer's suggestions.
>
> **why not call the search point "the current policy"? The same for "solutions", they are policies.**
>
> We have removed the term "solution" and replaced it with "policy" throughout the manuscript to avoid unnecessary confusion.
>
> **"some minimum threshold". Why call this a minimum threshold? Threshold for what? I rather see it as an exploration bonus...**
>
> In some cases, if the threshold value is set too low, the algorithm can initially fill the archive with very poor performing policies from which gradient arborescence cannot recover the true optimal policies. Intuitively, this can be thought of as seeding the archive with policies too far from the manifold and then attempting to perform search nowhere near the manifold. This min-threshold value prevents too poorly performing policies from entering the archive in the first place.
>
> **"in the direction of the archive that is least" -> doesn't this also take performance into account?**
>
> You are correct, although the ranking implicitly favors policies that land in new cells over those that improve on known ones. We have rephrased this to say "in the direction of greatest archive improvement"
>
> **In 2.4: "In prior work (ref) ... In this work" -> You should not write this in a way that let us know who you are.**
>
> We have done a pass through the introduction and related works sections of the paper and re-balanced the citations.
>
> **In 3.2, I'm afraid the claim that trust region methods provide formal guarantees on the quality of the gradient estimates is wrong.**
>
> While we agree that papers often make unrealistic assumptions to formalize proofs, this theory-practice gap is ubiquitous in nature, and present in other areas of AI and robotics research such as in robot safety, multi-agent planning and collision avoidance, etc. Rather than saying the proof of TRPO formally holds in the QD-RL setting, we wish to convey the idea that constrained optimization of the policy, especially in settings with non-stationary objective functions, is a nice property to have rather than not. We have softened the language in the manuscript to better convey this. "Being an approximate trust region method, the constrained policy update step provides some robustness to noisy and non-stationary objectives."

---

> > ### Author Response · Authors · 2023-11-16
> > **Response to Reviewer vTCA Part 2**
> >
> > **You use the MMPs as rewards to optimize. So it means that your algorithm is looking for as much leg contact as possible. Shouldn't it be looking for as much "leg contact diversity" as possible instead? This is unclear. Could you clarify?**
> >
> > That is correct, we estimate the gradient to increase foot contact time. However, during the branching phase, the gradient coefficients we sample are unconstrained. We could sample a range of positive and negative coefficients, resulting in a range of lower and higher foot contact times of varying degree. The onus is thus on the evolution strategy to update the search distribution such that we pick gradient coefficients that maximally improve the archive, which implies maximizing diversity.
> >
> > **"We additionally modify the computation of the policy gradient into a batched policy gradient method, where intermediate gradient estimates of each function w.r.t. policy params only flow back to the parameters corresponding to respective individual policies during minibatch gradient descent." -> This part is very unclear to me, a small diagram or equations would probably help.**
> >
> > While estimating the objective-measure Jacobian, we use a single, vectorized policy $\pi_{\theta}$ s.t. $\theta = <\theta^1, ..., \theta^{k+1}>$, where each of the $k+1$ parameter vectors correspond to parameters for separate behavior policies optimizing separate objectives. Since they correspond to different policies, we must slice the total number of environments $|E|$ into $k+1$ chunks of size $\frac{|E|}{k+1}$. Each i'th chunk will generate a batch of data  $B^i = (s_{t}, a_{t}, r_{t}, s_{t+1}, ..., s_{t+K})^i$, where $K$ is the rollout length. During the update step, the advantages and importance sampling ratios are only computed within each batch i.e. we do not mix data across batches. While performing gradient descent, the losses across batches are summed and backpropagated through the computation graph of the vectorized policy. However, since only the $i'th$ sub-policy $\pi_{\theta^i}$ was used to create the i'th batch of data $B^i$, only the gradients corresponding to the i'th loss term will flow back to these parameters inside of the vectorized policy. This essentially allows us to jointly optimize $\pi_{\theta^1}, ..., \pi_{\theta^{k+1}}$ without spawning separate PPO instances for each policy and sequentially computing the gradients of $f, m_{1}, ..., m_{k+1}$.
> >
> > **There are many considerations about using constant variance or not...**
> >
> > To the best of our knowledge, we have not seen examples of PPO used with a fixed, unlearnable standard deviation parameter for the action distribution. We would appreciate if the reviewer could provide references to this in the literature, and we would be happy to update the paper accordingly. TRPO and PPO indeed have options for state-dependent and state-independent *learnable* standard deviation parameters. However, we are concerned with using a fixed, unlearnable standard deviation parameter as opposed to a learnable one.
> >
> > In the canonical training paradigm, the standard deviation parameter decreases as the policy approaches optimality such that actions are sampled quasi-deterministically at convergence (many RL libraries provide the option to directly use the action means instead of sampling at test time). In the DQD setting, however, this is undesirable behavior. Ideally, the parameter would adaptively decrease or increase depending on if we are converging to a peak in the objective landscape or moving away from one. In our experiments, we have found that no matter the choice of hyperparameter, the addition of exploration bonuses, etc., that enabling gradient flow to this parameter results in it monotonically decreasing to 0. As this parameter converges, it becomes increasingly difficult to sample new actions and have positive advantages, which essentially cripples the branching and walking phase of PPGA since it relies on gradient estimates from PPO. We acknowledge that making this parameter static in some environments is a nonideal solution and would like to have a more theoretically grounded approach in future iterations of this work. However, in the absent of such a tool, we instead opted for this choice and only include it in the paper to be fully transparent in our design decisions.
> >
> > **About Section 3.3, I think the first paragraph which establishes that xNES is a better option could be moved in an appendix**
> >
> > We agree that section 3.3 contains unnecessary details and have simplified this to a single paragraph explaining the choice of switching from CMA-ES to xNES. We have also recovered space in other parts of the paper to move the TD3GA ablation to the main paper, as per the reviewer's suggestions.
> >
> > **The fact that there is such a learning rate is not explained before.**
> >
> > The archive learning rate is $\alpha$ referred to in the last paragraph of section 2.3 describing CMA-MAE/MAEGA. We have made this more explicit in the revised manuscript.

---

> > > ### Author Response · Authors · 2023-11-16
> > > **Response to Reviewer vTCA Part 3**
> > >
> > > **All experimental figures and tables make a poor use of space.**
> > >
> > > **In Fig. 4, using +/- one std as variance information is a bad practice in RL, where the variance is generally not Gaussian...**
> > >
> > > We have revised all figures and updated all plots to use bootstrapped confidence intervals to represent the shaded regions in accordance to the papers provided by the reviewer.
> > >
> > > **"We present a new method, PPGA, which is one of the first QD-RL methods to leverage on-policy RL," -- one of the first, so what are the others?**
> > >
> > > To the best of our knowledge, [1] and [2] are the only other concurrent works that leverage on-policy RL. However, [1] does not provide a reference implementation with PPO (only TD3) in their official repo, and [2] does not present any findings with on-policy RL, although it should, in theory, work with on-policy RL.
> > >
> > > [1] Wu S, Yao J, Fu H, et al. Quality-Similar Diversity via Population Based Reinforcement Learning. In The Eleventh International Conference on Learning Representations, (2023).
> > >
> > > [2] Pierrot, Thomas, and Arthur Flajolet. "Evolving Populations of Diverse RL Agents with MAP-Elites." arXiv preprint arXiv:2303.12803 (2023).
> > >
> > > **"We show that DQD algorithms and on-policy RL have emergent synergies that make them work particularly well with each other." -- I'm sorry but this point does not emerge clearly from reading the paper. You should have somewhere a paragraph about the investigations of these synergies.**
> > >
> > > In sections 2 and 3, we attempt to convey how on-policy RL is particularly well-suited for the DQD setting, and the specific design decisions made to exploit this synergy. In section 4, we show the resulting formulation yields state of the art results compared to baselines. The choice of ablations and baselines attempts to answer the counterfactuals "what if component X of PPGA were replaced with component Y?". For example, what if PPO were replaced with TD3 and all other components were held constant? To this end, we completely agree with the reviewer that certain comparisons such as PPGA vs TD3GA should be central to the paper, and have updated the manuscript accordingly. However, we believe these comparisons justify the claim about the synergies between on-policy RL and DQD.
> > >
> > > **Your paper needs slightly more than 9 pages. Again, it could have been desk-rejected for that. Please follow the rules.**
> > >
> > > All content of our paper fits within the 9 pages --- the section after our paper content is a reproducibility statement that does not count towards the page limit (author guidelines: https://iclr.cc/Conferences/2024/AuthorGuide). We apologize for any confusion.

---

> > > > ### Comment · Reviewer_vTCA · 2023-11-18
> > > > **Good job**
> > > >
> > > > After reading the other reviews and all the authors responses to all reviewes including mine, it seems to me that the authors have done a good job and that this time, their paper is in a good position for being accepted. I also checked the revised version and found it significantly better. Hence I'm happy to increase my score to a full "accept".
> > > >
> > > > Side note: in the introduction, a "non-markovian" should be replaced by "non-Markovian".

---

> > > > > ### Comment · Reviewer_vTCA · 2023-11-18
> > > > > **PS : non-stationarity and PPO**
> > > > >
> > > > > PS : I found the discussion with reviewer RPaL about non-stationarity and why PPGA needs the trust region property from the PPO clipping mechanism interesting in itself. Putting forward a study of this specific aspect together with dedicated experimental studies may deserve a side paper for itself, maybe for a workshop.

---

> > > > > > ### Author Response · Authors · 2023-11-19
> > > > > > **Response to Reviewer vTCA**
> > > > > >
> > > > > > We are very happy to hear that the reviewer has updated their score! We thank the reviewer again for their incredibly valuable feedback and insights, and agree that the paper is in a much better position as a result of the discussion. (We have also capitalized the "m" in "non-Markovian" and uploaded the revised pdf)
> > > > > >
> > > > > > **PS : I found the discussion with reviewer RPaL about non-stationarity and why PPGA needs the trust region property from the PPO clipping mechanism interesting in itself. Putting forward a study of this specific aspect together with dedicated experimental studies may deserve a side paper for itself, maybe for a workshop.**
> > > > > >
> > > > > > This would indeed be interesting to study! We are also interested in expanding PPGA's capabilities to discrete action spaces, such as for Atari games. A study on the importance of constrained optimization in various tasks outside of continuous control in the QD-RL setting is certainly relevant and would be an important result to report.

---

### Official Review · Reviewer_2hvw · 2023-10-31

**Soundness:** 3 good
**Presentation:** 2 fair
**Contribution:** 2 fair
**Rating:** 6
**Confidence:** 4

**Summary:**

This paper presents Proximal Policy Gradient Arborescence (PPGA), a novel algorithm that combines PPO with Differentiable Quality Diversity to improve the performance of QD-RL in challenging locomotion tasks. The authors propose several key changes to existing methods, including a vectorized implementation of PPO, generalizing CMA-based DQD algorithms using Natural Evolution Strategies, introducing Markovian Measure Proxies, and a new method for moving the search point in the archive. The experiment results show that PPGA achieves state-of-the-art results, with a 4x improvement in best reward over baselines on the challenging humanoid domain.

**Strengths:**

1. The paper addresses a significant gap in performance between QD-RL and standard RL algorithms on continuous control tasks. The results demonstrate state-of-the-art performance of PPGA, which combines the strengths of PPO and DQD, on challenging locomotion tasks, with a 4x improvement in best reward over baselines.
2. The paper provides a clear and well-structured presentation of the proposed algorithm and its components.

**Weaknesses:**

1. The paper could benefit from a more detailed comparison with other QD-RL methods and an explanation of why PPO was specifically chosen over other RL algorithms.
2. The generalizability of the proposed algorithm to other domains and tasks beyond locomotion is not discussed.
3. It would be better to see some discussion about the scalability of the proposed algorithm, especially in comparison to existing methods.

**Questions:**

1. Can you provide more insight into why PPO was specifically chosen over other RL algorithms for this work?
2. How does the proposed PPGA algorithm perform in other domains and tasks beyond locomotion?
3. Can you discuss the scalability of the proposed algorithm, especially in comparison to existing methods?

---

> ### Author Response · Authors · 2023-11-16
> **Response to Reviewer 2hvw**
>
> Thank you for your feedback. We will respond to each concern individually:
>
> **Can you provide more insight into why PPO was specifically chosen over other RL algorithms for this work?**
>
> Regarding other QD-RL methods, we selected recent methods that have demonstrated high performance and have accessible open-source implementations. In particular, we chose PGA-ME, the current state-of-the-art, as well as QD-PG, separable CMA-MAE, and CMA-MAEGA (TD3, ES), all of which have shown competitive performance in prior work. Appendix H also includes a comparison to PBT-ME (SAC), which we only ran on Humanoid due to computational expense. We believe these baselines are representative of the current state of the QD-RL field.
>
> Regarding PPO, we believe PPO is an ideal choice for integration into QD-RL algorithms for several reasons. First, PPO can leverage the high parallelization of GPU-accelerated simulators such as Brax to significantly accelerate learning. Current QD-RL methods rely on off-policy RL algorithms such as TD3 and SAC, which are designed to be sample-efficient and benefit from faster sequential execution. In other words, these off-policy methods operate with minimal amounts of data. We are interested in the other end of this spectrum -- that is, when large amounts of data are available, what kind of methods benefit the most? There is a large body of evidence that shows PPO continues to be competitive in these regimes even on contemporary robot learning problems [1, 2, 3]. This was a significant factor when choosing which RL algorithm to investigate.
>
> Finally, regarding our choice of PPO over other on-policy methods such as IMPALA, PPO's use of an approximate trust region to constrain the policy update empirically provides robustness to noisy and even non-stationary objectives. This is especially important in the QD-RL setting, where the branching phase and walking the search policy require optimizing non-stationary objectives.
>
> [1] Rudin, Nikita, et al. "Learning to walk in minutes using massively parallel deep reinforcement learning." Conference on Robot Learning. PMLR, 2022.
>
> [2] Handa, Ankur, et al. "DeXtreme: Transfer of Agile In-hand Manipulation from Simulation to Reality." arXiv preprint arXiv:2210.13702 (2022).
>
> [3] Batra, Sumeet, et al. "Decentralized control of quadrotor swarms with end-to-end deep reinforcement learning." Conference on Robot Learning. PMLR, 2022.
>
> **How does the proposed PPGA algorithm perform in other domains and tasks beyond locomotion?**
>
> For this paper, we focused on comparing the algorithmic design choices of PPGA and the resulting performance differences against prior SOTA QD-RL algorithms. As such, we ran PPGA on benchmarks that were introduced in [1] and have become standard in the field for a majority of QD-RL methods.
> However, we are interested in applying PPGA to non-standard QD tasks with discrete action spaces such as Atari.
> Given the prior success of PPO in a wide variety of tasks with discrete and continuous actions spaces, we believe PPGA has a good chance of succeeding in these tasks.
>
> [1] Olle Nilsson and Antoine Cully. 2021. Policy gradient assisted MAP-Elites. In Proceedings of the Genetic and Evolutionary Computation Conference (GECCO '21). Association for Computing Machinery, New York, NY, USA, 866–875. https://doi.org/10.1145/3449639.3459304
>
>  **Can you discuss the scalability of the proposed algorithm, especially in comparison to existing methods?**
>
> We found empirical evidence of scalability of our method from two perspectives: scaling to larger archives and learning from more data.
> We present results in scaling to larger 50x50 archives and compare to baselines in Appendix G.
> PPGA scales well and maintains state of the art performance over baselines when archives of thousands of policies are concerned.
> We believe this is a direct result of the fact that, like prior DQD-based algorithms, PPGA *explicitly* estimates the gradient of greatest archive improvement via gradient arborescence.
> That is, the information on what new cells were discovered during the branching phase is utilized by xNES to compute the natural gradient w.r.t. archive improvement.
> Critically, the higher resolution the archive i.e. the more cells there are, the better this gradient estimate is, and so PPGA may in fact favor larger archives.
>
> Finally, we demonstrate PPGA's ability to continually learn with more data by running an additional experiment where PPGA and the current state of the art baseline PGA-ME were run for 1.2 million evaluations, or twice as long as the experiments in the main paper, on larger 50x50 archives. The results have been added to Appendix G. We notice that the average performance of the archive, i.e., the average performance across all policies in the archive, continues to increase, suggesting that PPGA continuously optimizes for performance after the initial discovery phase. We observe this effect to a much lesser extent with PGA-ME.

---

> > ### Comment · Reviewer_2hvw · 2023-11-23
> >
> > Thanks for your response.  I am happy that the authors solve most of my questions. This paper is good enough to be accepted from my perspective.  It is also interesting to try these methods in Atari, in which most of the games have a discrete action space and are more complex than MuJoCo. It would be nice to compare QD-RL methods on Atari in the future version.

---

### Official Review · Reviewer_1LjB · 2023-11-01

**Soundness:** 3 good
**Presentation:** 3 good
**Contribution:** 2 fair
**Rating:** 6
**Confidence:** 3

**Summary:**

A RL policy can learn to maximize cumulative reward in an environment by discovering an optimal policy. This paper shows that there are potentially many different sets of behaviors that could achieve optimal or near optimal performance. For example the authors highlight that in the humanoid walker environment different strategies can be discovered that all perform well: jumping, galloping, walking... These strategies can all be characterized by some measurements that are computed on trajectories. Figure 1 demonstrates this concept quite well. It plots two different measures with a heatmap of how well the policy at each point fulfilled the main objective (locomotion)

There are already prior works which establish this idea of characterizing policies with measures and attempting to maximize diversity in measure space. There are works that focus on differentiable measures and works that use non differentiable measures. This paper considers non-differentiable measures and approximates them using the bellman equation.

Several value functions are learned which each approximate the instantaneous measure functions (deltas) and the overall objective function f. ***Each measure function is the average of the instantaneous measurement (delta) in each state.***

Since all value functions are differentiable then a gradient of policy parameters with respect to a linear combination of these value functions can be obtained. Different weights can be assigned to each value function to cause the actor update to move in a different direction in the archive / measure space.

**Strengths:**

Overall this is an interesting work. Massive (4x) performance gain compared to baselines.
Nice interpretability of archive exploration. The archives produced by PPGA seem more
interesting to look at. There are more distinct peaks shown in Fig 3 than the baseline.

**Weaknesses:**

I would suggest the authors make the explanation of the policy sampling more clear since that is what confused me the most.

The notation for the gradient of the measures (\nabla m) is not clear. Is it the gradient of the measure with respect to the policy parameters? (this is explained only later in the paper)

I suggest moving pseudocode Algorithm 1 to the main text.

**Questions:**

CMA-ES is introduced but not really elaborated upon. It seems to be very important in sampling different policy parameters. how is branching the policy parameters in different directions in measure space done?

looks like Section 3.3 contains unnecessary formalism which is not used elsewhere in the paper. why is it provided?

---

> ### Author Response · Authors · 2023-11-16
> **Response to Reviewer 1LjB**
>
> Thank you for your feedback and questions. We will respond to each point individually:
>
> **I would suggest the authors make the explanation of the policy sampling more clear since that is what confused me the most.**
>
> Our sampling method builds upon that of CMA-MAEGA described in Sec. 2.3. In summary, CMA-MAEGA branches off new policies from the current one by sampling *gradient coefficients*, linearly combining them with the objective-measure gradients in various ways, and applying these gradients to the current policy. A single gradient arborescence starts at the current policy $\pi_{\theta_\mu}$ and adds a linear combination of objective and measure gradients to parameters $\theta_\mu$. Formally, this linear combination is $c_0 \nabla f + \sum_{j=1}^k c_j \nabla m_j$, where $\textbf{c} = c_{0..k}$ are the sampled gradient coefficients. In CMA-MAEGA, multiple vectors of coefficients are sampled simultaneously, resulting in multiple branches. The gradient coefficients are sampled from a gaussian search distribution with mean $\mu$ and covariance $\Sigma$; $\mu$ and $\Sigma$ are updated by CMA-ES in the case of CMA-MAEGA, and xNES in the case of PPGA. Intuitively, sampling many coefficients produces multiple branched policies, increasing the likelihood of discovering new and higher performing behaviors. $\mu, \Sigma$ biases the branching in the direction that is most likely to lead to exploring new parts of the archive. We have updated the language in section 2.3 to use terms more familiar to the standard machine learning audience and hope that this brings additional clarity to the sampling procedure.
>
> **The notation for the gradient of the measures ($\nabla m$) is not clear. Is it the gradient of the measure with respect to the policy parameters? (this is explained only later in the paper)**
>
> That is correct. By measure gradient, we specifically mean $\frac{\partial m_i}{\partial \theta}$ i.e. the gradient of the i'th measure w.r.t. policy params, since there can be multiple measure functions. We have added this clarification to the beginning of section 2.3.
>
> **I suggest moving pseudocode Algorithm 1 to the main text.**
>
> We agree that moving Algorithm 1 to the main text would improve clarity; however, our main obstacle is fitting it within the page limit. For the final version, we will attempt to trim content and insert Algorithm 1, but it may not be possible (the final version also has 9 pages, identical to the submission version).
>
> **CMA-ES is introduced but not really elaborated upon. It seems to be very important in sampling different policy parameters. how is branching the policy parameters in different directions in measure space done?**
>
> We would like to clarify that CMA-ES is only employed by our predecessor method CMA-MAEGA, and is replaced by xNES in PPGA, a functionally equivalent method with more theoretically grounded components discussed in Section 3.3. CMA-ES is a natural evolution strategy that has traditionally been used as a black-box optimizer of solution parameters for non-differentiable objective functions. In DQD papers such as CMA-MAEGA, it instead is employed to maintain and update a search distribution in objective-measure *gradient coefficient space* such that the solutions created during the branching phase have the highest likelihood of achieving novel and high performing behaviors. In PPGA, CMA-ES was replaced with another natural evolution strategy, xNES, due to the training instabilities induced by CMA-ES on high dimensional RL domains. We refer the reader to [1] for a detailed review on the challenges of employing CMA-ES on RL tasks, and to the answer above on the specific details of branching in measure space.
>
> [1] Müller, Nils, and Tobias Glasmachers. "Challenges in high-dimensional reinforcement learning with evolution strategies." Parallel Problem Solving from Nature–PPSN XV: 15th International Conference, Coimbra, Portugal, September 8–12, 2018, Proceedings, Part II 15. Springer International Publishing, 2018.
>
> **looks like Section 3.3 contains unnecessary formalism which is not used elsewhere in the paper. why is it provided?**
>
> We will assume the reviewer is referring to the description of natural evolution strategies in the second paragraph of Sec. 3.3. We initially added this formalism to help describe how natural evolution strategies operate, but we agree that it is not necessary for providing insight. We have removed the unncessary formalism and simplified this section into a single paragraph.

---

### Official Review · Reviewer_RPaL · 2023-11-01

**Soundness:** 3 good
**Presentation:** 3 good
**Contribution:** 3 good
**Rating:** 8
**Confidence:** 2

**Summary:**

The authors propose Proximal Policy Gradient Arborescence (PPGA), a new QD-RL algorithm that builds on a DQD algorithm, specifically CMA-MAEGA, with an on-policy RL algorithm, specifically PPO, as the gradient estimator. To make this integration more stable, the authors propose multiple changes: 1) replace the CMA-ES in CMA-MAEGA with a more contemporary xNES algorithm to update the distributions, as a solution to accommodate the noisy RL objective, 2) use PPO with weighted rewards to walk the search point to avoid gradient staleness. The authors also implement a vectorized PPO for efficient gradient estimation.

Empirically, the author demonstrates that PPGA achieves much higher quality solutions compared with previous QD-RL baselines with matching coverage rates in multiple Mujoco locomotion domains.

**Strengths:**

- The paper is generally well-written and easy to follow.
- The proposed techniques to accommodate the integration issues of PPO are all logically and empirically justified.
- There are substantial empirical improvements over previous baselines, especially in terms of solution qualities. As a side note, I do find the introduction of the Complementary Cumulative Distribution Function provides a very intuitive way of evaluating the solution quality and diversity.

**Weaknesses:**

- At the beginning of Sec 3.2, the authors claim "Being an approximate trust region method, it provides some formal guarantees on the quality of the gradient estimates". Could the authors expand on this and more formally establish this claim? This claim is an important reason of choosing PPO since there are other parallelizable RL algorithm choices such as IMPALA[1].
- I have a slight concern that all the ablation studies show a dramatic decrease in performance. Could finding a better set of hyperparameters instead of inheriting the PPGA hyperparameters help?

[1] Espeholt, Lasse et al. "IMPALA: Scalable Distributed Deep-RL with Importance Weighted Actor-Learner Architectures." Proceedings of the International Conference on Machine Learning (ICML). 2018.

**Questions:**

I don't have more questions apart from the ones in the weaknesses section.

---

> ### Author Response · Authors · 2023-11-16
> **Response to Reviewer RPaL**
>
> Thank you for your comments. We will respond to each of the reviewer's concerns individually:
>
> **At the beginning of Sec 3.2, the authors claim "Being an approximate trust region method, it provides some formal guarantees on the quality of the gradient estimates". Could the authors expand on this and more formally establish this claim? This claim is an important reason of choosing PPO since there are other parallelizable RL algorithm choices such as IMPALA.**
>
> PPO and IMPALA both indeed have similar derivations in that they use importance sampling (IS) when computing the policy update objective: $r_t(\theta) = \left[ \frac{\pi_{\theta}(a|s)}{\pi_{\theta_{old}}(a|s)} \hat{A}_t \right]$,
>
>  where $\pi_{\theta_{old}}$ is some behavior policy $\pi_w$ in the case of IMPALA. Critically, the constrained update of $r_t(\theta)$ between $1 - \epsilon$ and $1 + \epsilon$ in PPO that IMPALA omits matters in two locations of the PPGA algorithm: (1) when we are estimating the objective-measure Jacobian during the branching phase, and (2) when we are walking the current search policy according to the multi-objective reward function. This is mainly due to the fact that the objectives are non-stationary in both cases.
> Consider, for example, two subsequent QD iterations $i-1, i$. On iteration $i$, during the later phase of walking the current policy, we optimize $\pi_{\theta}^{i-1}(a|s), V^{i-1}_{\phi}$
>
> according to the multi-objective reward function $r^i_{multi} = <c^i_{\mu_0}, ..., c^i_{\mu_k}> \cdot <f, \delta_0, ..., \delta_k>$ for $N_2$ steps, giving us $\pi_{\theta}^{i}(a|s), V^{i}_{\phi}$.
>
> In the canonical RL training paradigm, we optimize towards a single objective $r_{static}$ such that the MDP is ergodic and the state visitation distribution $p(s)$ is stationary. In this setting, a case can be made for unconstrained optimization of the (IS) objective. However, consider the adversarial case in PPGA when the multi-objective reward coefficients between subsequent iterations are orthogonal $<c_{\mu_1}, ..., c_{\mu_k}>^{i-1} \cdot <c_{\mu_1}, ..., c_{\mu_k}>^{i} = 0$, for example when the least explored direction is to the left on iteration $i-1$ and then up on iteration $i$ in a 2D archive. Then, $p(s)$ becomes highly non-stationary, and unconstrained updates to $\pi_{\theta}$ can lead to visiting very new states $s'$ in which $V_{\phi}$ makes poor predictions, resulting in oscillations and destabilizing training. Trust region methods such as PPO prevent us from making too large updates on the policy too quickly, allowing $V_{\phi}$ to ingest new states slowly and provide stable advantage estimates for the policy update step.
>
> As reviewer vTCA pointed out, many formal guarantees made by the TRPO paper make assumptions that are often violated in practice, and this is certainly the case in the QD-RL setting. As such, we have softened the language to instead read "Being an approximate trust region method, the constrained policy update step provides some robustness to noisy and non-stationary objectives". We still maintain the belief that constrained optimization is a property worth having in this setting rather than not.
>
> **I have a slight concern that all the ablation studies show a dramatic decrease in performance. Could finding a better set of hyperparameters instead of inheriting the PPGA hyperparameters help?**
>
> In the xNES vs CMA-ES ablation, we made two attempts using two separate, highly-tuned and optimized reference implementations of CMA-ES inside PPGA provided by [1] and [2]. We noticed the training divergence in both instances across multiple seeds.
>
> Regarding the TD3GA ablation, we use a reference TD3 implementation used by other off-policy QD-RL papers such as PGA-ME for QD-RL tasks. Admittedly, certain hyperparameters such as the optimal size and update schedule of the replay buffer in high-throughput regimes remains an open question that could be further optimized. However, given that using TD3 as a drop-in replacement for PPO in PPGA leads to a fundamentally different learning paradigm, we argue that this algorithm and an involved hyperparameter search warrants its own proper investigation as a separate line of work.
>
> [1] Tjanaka, Bryon, et al. "pyribs: A bare-bones Python library for quality diversity optimization." arXiv preprint arXiv:2303.00191 (2023).
>
> [2] Nikolaus Hansen, Youhei Akimoto, and Petr Baudis. CMA-ES/pycma on Github. Zenodo, DOI:10.5281/zenodo.2559634, February 2019.

---

> > ### Comment · Reviewer_RPaL · 2023-11-20
> >
> > Thank you for the response. I believe my concerns have been addressed.

---

### Meta-Review · Area_Chair_btgz · 2023-12-05

**Metareview:**

Good paper, improving previous results in this field. The reviewers' concerns have been addressed during the discussion period and they all recommend acceptance.

**Justification For Why Not Higher Score:**

While the paper has some novel ideas, it builds upon previous techniques in a relatively narrow field. I feel that a spotlight score is appropriate given the reviews.

**Justification For Why Not Lower Score:**

The paper makes none trivial contributions, develops a relatively new technique for QD in RL, and has strong empirical results.

---

### Decision · Program_Chairs · 2024-01-16

Accept (spotlight)